# LiveMoments: Reselected Key Photo Restoration in Live Photos via Reference-guided Diffusion

**Clara Xue**[1*]   **Zizheng Yan**[1*]   **Zhenning Shi**[1,2]   **Yuhang Yu**[1]   **Jingyu Zhuang**[1]
**Qi Zhang**[1]   **Jinwei Chen**[1]   **Qingnan Fan**[1†]
[1]vivo BlueImage Lab    [2]College of Computer Science, Nankai University
{clara.x0277,fqnchina}@gmail.com   zizhengyan@link.cuhk.edu.cn

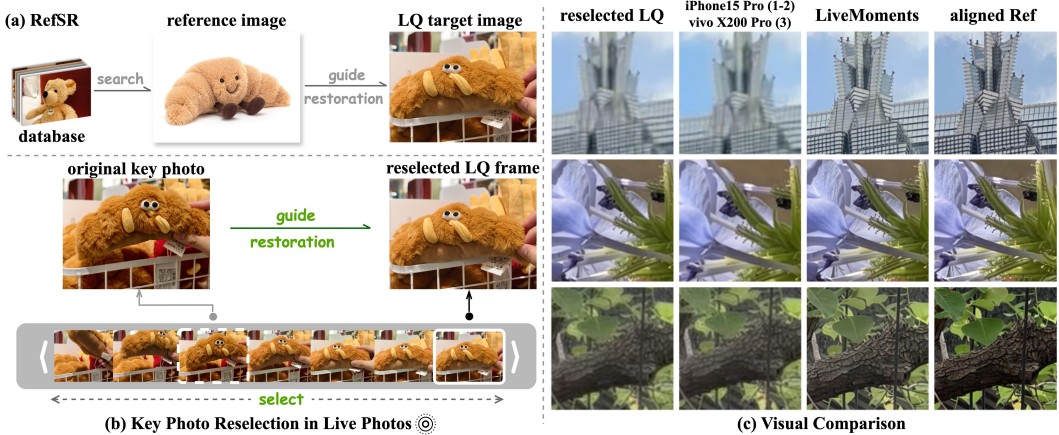

Figure 1: Illustration of **Reselected Key Photo Restoration in Live Photos** and visual comparison. While RefISR adopts external reference image with only semantic similarity, our setting leverages both the reference and target images from the same Live Photo sequence, ensuring a shared temporal context. The proposed **LiveMoments** significantly outperforms the premium smartphones.

## Abstract

Live Photo captures both a high-quality key photo and a short video clip to preserve the precious dynamics around the captured moment. While users may choose alternative frames as the key photo to capture better expressions or timing, these frames often exhibit noticeable quality degradation, as the photo capture ISP pipeline delivers significantly higher image quality than the video pipeline. This quality gap highlights the need for dedicated restoration techniques to enhance the reselected key photo. To this end, we propose *LiveMoments*, a reference-guided image restoration framework tailored for the reselected key photo in Live Photos. Our method employs a two-branch neural network: a reference branch that extracts structural and textural information from the original high-quality key photo, and a main branch that restores the reselected frame using the guidance provided by the reference branch. Furthermore, we introduce a unified Motion Alignment module that incorporates motion guidance for spatial alignment at both the latent and image levels. Experiments on real and synthetic Live Photos demonstrate that *LiveMoments* significantly improves perceptual quality and fidelity over existing solutions, especially in scenes with fast motion or complex structures.

---

*Equal contribution. †Corresponding author.

# 1 INTRODUCTION

Unlike traditional photographs that capture a static frame, a Live Photo preserves the fleeting moments around a shutter click. To achieve this, each Live Photo consists of two components: (a) a high-quality (HQ) key photo[1] taken at the capture moment and (b) a low-quality (LQ) video clip approximately 3 seconds, spanning moments around the key photo. This format not only enables dynamic visual recording, but also offers the flexibility for users to reselect a preferred frame as the new key photo. However, these reselected frames often exhibit significantly reduced quality. While the original key photo is processed through the complete ISP pipeline with advanced enhancements, the alternative frames are extracted from a compressed, low-latency preview stream and are further degraded by motion blur and sensor noise. To ensure optimal user experience, it is essential to restore the quality of the reselected frame to match that of the original key photo, while preserving content fidelity.

To this end, we introduce a new task: **Reselected Key Photo Restoration in Live Photos**, where the original key photo serves as a reference to guide the restoration of the reselected frame. We position this task as a sub-category of Reference-based Super-Resolution (RefSR), with a unique setting that restores a single LQ frame guided by a single reference from the same Live Photo sequence. Unlike conventional Reference-based Image Super-Resolution (RefISR), which restores the LQ image with an HQ reference from external databases, our task leverages an in-sequence reference that ensures content consistency. Meanwhile, Reference-based Video Super-Resolution (RefVSR) enhances full video sequences but struggles with high-resolution inputs (*i.e.*, 4K), and methods are typically built on triple-camera smartphone datasets, with videos from different fields of view recorded simultaneously. In contrast, our task focuses on real Live Photos, where a single 4K frame is restored with guidance from a temporally offset reference. This setting naturally involves more dynamic real-world scenes, making it novel and practical while remaining efficient in both time and memory. Furthermore, our task introduces unique challenges, including significant quality gap and motion misalignment between the reselected frame and the original key photo, due to subject movement or camera shake.

These challenges impose limitations on existing RefSR methods for the task of reselected key photo restoration. Traditional RefISR methods, typically based on CNNs or transformers (Zhang et al., 2019; Jiang et al., 2021), adopt various feature matching strategies but are constrained by relatively small model size and the lack of strong pre-trained priors, making them insufficient for handling the diverse degradations and motion misalignments in Live Photos. Although diffusion-based RefISR methods demonstrate stronger generative capabilities, only two such methods have been explored to date and often produce unnatural textures. ReFIR (Guo et al., 2024) relies on a fixed-coefficient gating mechanism with limited robustness, while CoSeR (Sun et al., 2024) generates HQ references from CLIP-based embeddings that neglect local detail alignment and fuses features by prioritizing LR content, making it less suitable for reference-driven tasks. Beyond image-based settings, RefVSR methods remain confined to traditional designs and exploit full sequence (Lee et al., 2022) or single middle-frame (Kim et al., 2023) references with temporal propagation. Such designs are utilized to handle small misalignments but fail under the larger temporal offsets and quality gaps in Live Photos. In addition, single image SR (SISR) methods (Wang et al., 2024; Wu et al., 2024b) overlook the reference and often fail to preserve accurate structure and details in the presence of motion.

Therefore, we propose LiveMoments, a diffusion-based framework tailored for reselected key photo restoration in Live Photos. It leverages diffusion priors for fine-grained feature extraction and employs attention-based fusion to guide the RestorationNet in selectively incorporating well-aligned reference features within a shared feature space, enabling precise and targeted reference-driven conditioning. To address motion misalignment between the two inputs, we further introduce a unified Motion Alignment module that operates at both the latent and image levels. At the latent level, we propose a *motion-guided attention* that injects spatially aligned guidance into the latent space for more coherent feature fusion. At the image level, we design a *patch correspondence retrieval* strategy that captures patch-wise motion to locate the corresponding reference patches for consistent restoration in ultra-high resolution frames. To facilitate fair evaluation, we introduce a comprehensive benchmark for this task, including a synthetic dataset, SynLive260, and two real-world Live Photo datasets, vivoLive144 and iPhoneLive90, captured by consumer smartphones across diverse scenes. Furthermore, we adapt the no-reference metrics originally used in SISR and image generation to better suit our setting, where an HQ reference is available for evaluating result quality.

---

[1]From Apple's official documentation: `https://support.apple.com/en-sg/104966`

Our contributions can be summarized as follows:

- To the best of our knowledge, we are the first to address the problem of reselected key photo restoration in Live Photos. We propose LiveMoments, a diffusion-based framework tailored for this task, which leverages a dual-branch neural network with advanced feature fusion.

- To mitigate spatial misalignment in Live Photos, a unified Motion Alignment module is introduced to inject the motion guidance at the latent level while ensuring image-level content consistency.

- We establish a comprehensive benchmark consisting of three datasets and task-specific metrics. Extensive experiments show that LiveMoments significantly outperforms state-of-the-art RefSR and SISR methods in both quantitative and visual results, even under challenging real-world scenarios.

## 2 RELATED WORK

### 2.1 DIFFUSION-BASED SINGLE IMAGE SUPER-RESOLUTION (SISR)

Diffusion-based SISR restores the HQ images from a single LQ input via diffusion models, with recent research focusing on complex and unknown degradations. Building on the remarkable generative capabilities of diffusion models demonstrated in text-to-image (T2I) tasks (Rombach et al., 2022; Podell et al., 2023), recent works (Wang et al., 2024; Lin et al., 2024; Wu et al., 2024b; Yu et al., 2024) adapt these pretrained backbones to SISR by leveraging control mechanisms (*e.g.*, ControlNet (Zhang et al., 2023b)). StableSR (Wang et al., 2024) fine-tunes a time-aware encoder with the controllable feature warping module, while SeeSR (Wu et al., 2024b) uses degradation-aware text prompts. SUPIR (Yu et al., 2024) scales up generation by integrating large diffusion backbones with high-capacity adapters and datasets. CoSeR (Sun et al., 2024) utilizes a pre-trained T2I model to generate HQ references from LR embeddings for guided restoration. More recently, one-step distillation techniques are adopted to significantly reduce the number of diffusion steps by directly initializing from the LQ input. OSEDiff (Wu et al., 2024a) utilizes VSD loss (Wang et al., 2023b) as a regularization term, while TSD-SR (Dong et al., 2024) distills a multi-step SD3 model (Esser et al., 2024) into a one-step SISR solution via target score distillation. Despite these advances, most of the existing methods rely solely on generative priors and may produce visually plausible yet inaccurate content that sacrifice fidelity for perceptual richness. Such methods often fail to produce accurate results in real-world scenarios, where effectively preserving the original visual content is essential.

### 2.2 REFERENCE-BASED SUPER-RESOLUTION (REFSR)

RefSR aims to enhance LR inputs by leveraging external HR references. Image-based RefSR (RefISR) addresses single-image restoration and primarily focus on establishing accurate correspondence between the LR input and the reference image, enabling effective texture transfer and detail refinement. Early methods explore various matching strategies, including feature warping (Zheng et al., 2018), patch-level matching (Zhang et al., 2019; Yang et al., 2020), and multi-reference fusion (Zhang et al., 2023a). In addition, SSEN (Shim et al., 2020) employs deformable convolution (Dai et al., 2017) for adaptive feature alignment. Building on this, C2-Matching (Jiang et al., 2021) introduces a contrastive correspondence network with teacher-student correlation distillation, while DATSR (Cao et al., 2022) integrates deformable convolution with the Swin Transformer for enhanced performance. Recently, ReFIR (Guo et al., 2024) proposes a retrieval-enhanced architecture built on diffusion-based SISR models, achieving reference-guided texture propagation without additional training. Video-based RefSR (RefVSR) extends reference-based techniques to video super-resolution on smartphone datasets, where triple-camera systems simultaneously record videos from different fields of view. By leveraging consecutive frames, these methods integrate multi-frame information with reference to enhance restoration, with only minimal time gaps between frames and the reference. Methods such as RefVSR (Lee et al., 2022) and RefVSR++ (Zou et al., 2025) employ the entire HQ video sequence as reference and exploit bidirectional propagation to align multi-frame information, while ERVSR (Kim et al., 2023) improves efficiency by leveraging only a single reference frame. While our setting can be regarded as a sub-category of RefSR, it departs from conventional paradigms in both dataset construction and task formulation. To address the unique challenges posed by Live Photos, we construct dedicated datasets and design task-specific architectures.

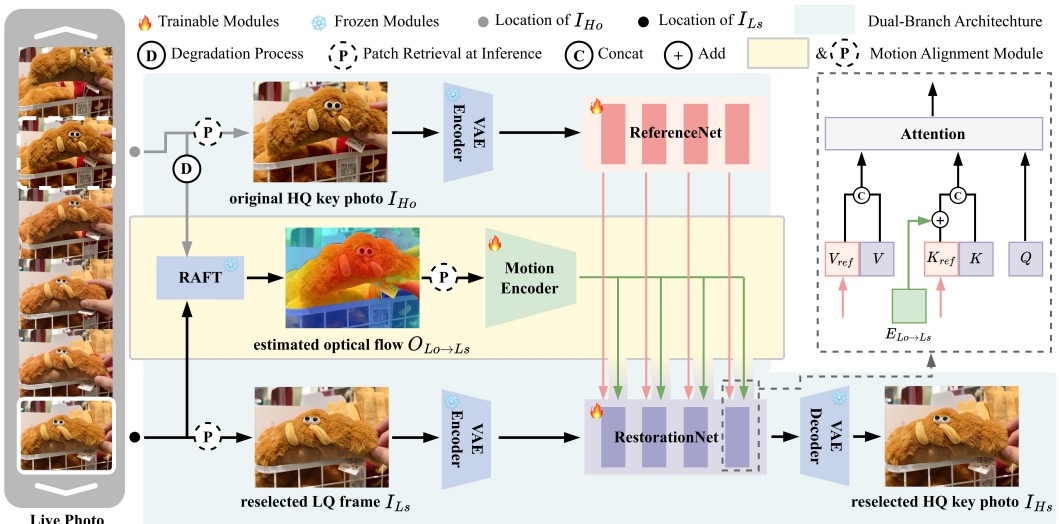

Figure 2: Overall architecture of LiveMoments. After the fixed VAE encoder, the original key photo and the reselected LQ frame are fed into the ReferenceNet and RestorationNet, respectively, and fused via cross-attention. For latent-level motion alignment, the optical flow is estimated with a fixed RAFT model and encoded with a Motion Encoder, which is further injected into the cross-attention as an additive bias. For image-level alignment, Patch Correspondence Retrieval (PCR) strategy is adopted during the inference to ensure spatial consistency when using the tiling strategy.

## 3 METHOD

### 3.1 PRELIMINARY

Given a Live Photo, we denote the reselected LQ frame as $I_{Ls} \in \mathbb{R}^{H \times W \times 3}$ and the original HQ key photo as $I_{Ho} \in \mathbb{R}^{H \times W \times 3}$. The goal of our task is to reconstruct a high-quality version of the reselected frame, denoted as $\widetilde{I}_{Hs}$, that matches the visual quality of $I_{Ho}$ while preserving the content fidelity of $I_{Ls}$. Thus, the task is formulated as learning a restoration model $G_\theta$ parameterized by $\theta$, which takes $(I_{Ls}, I_{Ho})$ as input and predicts $\widetilde{I}_{Hs}$.

Since real-world Live Photos lack HQ ground-truth counterparts for reselected frames, we construct the training dataset using HQ video sequences. For example, the ground-truth $I_{Hs}$ of the reselected frame and the original key photo $I_{Ho}$ are extracted with a clear temporal gap. $I_{Hs}$ is degraded to produce the LQ reselected frame $I_{Ls}$ (see Section 4.1 for more details). We employ flow matching (Albergo & Vanden-Eijnden, 2022; Esser et al., 2024) to train the diffusion model. Flow matching aims to transform Gaussian noise into a target distribution by learning an appropriate velocity field. In particular, Rectified Flow defines the forward process as $x_t = \alpha(t)x_0 + \beta(t)\epsilon$, where $\alpha(t) = 1 - t$ and $\beta(t) = t$. The training objective is $\mathbb{E}_{t,x_t} \|f(x_t) - dx_t/dt\|_2^2$, where $f$ denotes the neural network that parameterizes the velocity field in the Rectified Flow. To adapt flow matching to image-to-image translation tasks, several approaches condition the generation process on a source image, aiming to synthesize the target image from Gaussian noise. However, these methods often suffer from issues such as hallucinated textures. Inspired by bridge matching (Chadebec et al., 2025; Liu et al., 2023; Shi et al., 2023) that directly learns the velocity field between the source and target distributions, we leverage an objective that focuses on learning the velocity field between $I_{Ls}$ and $I_{Hs}$. Specifically, we define the forward process as:

$$z_t = \alpha(t)z_{Hs} + \beta(t)z_{Ls} + \sigma(t)\epsilon, \tag{1}$$

where $z_{Hs}$ and $z_{Ls}$ are the latent representations obtained by a VAE encoder in the setting of latent diffusion. The objective is to learn the velocity field:

$$\mathcal{L}_\theta = \mathbb{E}_{t,z_t} \|G_\theta(z_t, t) - \frac{dz_t}{dt}\|_2^2. \tag{2}$$

The details of $\alpha(t), \beta(t), \sigma(t)$ can be found in the *supplementary material*.

## 3.2 Overview of LiveMoments

Our setting is characterized by temporal dependency and reference-target correlation. Specifically, the reference and degraded frames are sampled from the same Live Photo sequence with a clear temporal offset, yet they retain scene-level coherence compared to static or externally sourced references. Consequently, effective restoration of the reselected key photo requires leveraging fine-grained details from the reference frame. To achieve this, the proposed LiveMoments consists of three key components, as shown in Fig. 2: a RestorationNet performs conditional denoising on the noisy latent of $I_{Ls}$; a ReferenceNet encodes the original key photo $I_{Ho}$ to provide high-quality guidance; a unified Motion Alignment module achieves fine-grained alignment at both the latent and image levels.

## 3.3 Dual-Branch Architecture

Inspired by the success of reference-guided generation in video tasks (Hu, 2024; Xu et al., 2024), we adopt a similar dual-branch design for reference-based restoration, leveraging both the diffusion priors and the attention mechanisms of the pre-trained T2I diffusion models. Unlike CLIP-based structure (Radford et al., 2021), which focuses on global semantics for low-resolution inputs, the ReferenceNet preserves high-resolution features and fine-grained details that are essential for high-fidelity restoration. Structurally, it mirrors the denoising backbone, enabling weight initialization from pre-trained checkpoints and better feature alignment with the RestorationNet. In our implementation, both branches are built on the Stable Diffusion 3 (SD3) architecture (Esser et al., 2024), which provides a powerful generation backbone. The original key photo $I_{Ho}$ is first encoded into the latent space by a frozen VAE encoder, then processed by a DiT-based ReferenceNet to obtain detailed features. These features are integrated into the main branch through cross-attention, where the key and value from both branches are concatenated as follows:

$$\text{Cross\_attn} = \text{Softmax}\left(\frac{Q[K, K_{\text{ref}}]^\top}{\sqrt{d}}\right)[V, V_{\text{ref}}], \tag{3}$$

where $Q$ is the query from the attention layer of the RestorationNet, $K_{\text{ref}}$ and $V_{\text{ref}}$ are the key and value from the ReferenceNet, while $K$ and $V$ are those from the RestorationNet. $[\cdot, \cdot]$ denotes the concatenation operation, and $d$ is the channel dimension. This interaction enables the model to adaptively select and transfer textures and structural information from the reference image rather than relying solely on coarse semantic alignment.

## 3.4 Motion Alignment Module

### 3.4.1 Latent space motion alignment

While the dual-branch structure enables implicit feature matching and detailed information transfer, it is often insufficient for the task of reselected key photo restoration on Live Photos. The reselected frame $I_{Ls}$ typically suffers from motion blur and subject displacement, making it difficult to establish accurate correspondences and align with the original key photo. Moreover, the significant quality gap between the HQ key photo and the degraded reselected frame further hinders effective feature alignment and leads to unreliable fusion.

To address these challenges, we design a *motion-guided attention* that introduces explicit motion guidance into the latent space. In Live Photos, where temporal dependency naturally exists, optical flow serves as an intuitive mechanism for establishing spatial correspondence. Distinct from previous flow-based restoration methods, we transform the estimated flow into motion embeddings and incorporate them into the cross-attention, thereby providing alignment priors to guide attention toward relevant regions. Specifically, we utilize a pre-trained RAFT model (Teed & Deng, 2020) to estimate the optical flow $O_{Lo \to Ls}$, which serves as a dense pixel displacement field between the degraded original key photo $I_{Lo}$ and the reselected LQ frame $I_{Ls}$. During training, $I_{Lo}$ is synthesized by applying the same degradation parameters as those used for the reselected frame $I_{Ls}$. At inference time, since $I_{Ls}$ suffers from real-world degradation in Live Photos, we simulate the corresponding degradation on $I_{Ho}$ to narrow the quality gap and obtain more reliable motion estimation. To encode the estimated motion, we introduce a lightweight Motion Encoder with convolutional layers and SiLU activations that transforms the raw flow field into motion embeddings. These embeddings are

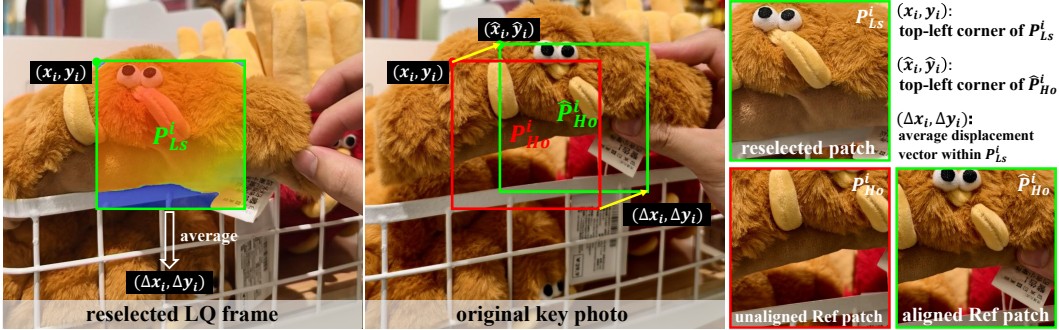

Figure 3: Illustration of the proposed Patch Correspondence Retrieval (PCR) strategy. The average displacement within a patch (the dense displacement field contained in $P_{Ls}^i$) is used to shift the top-left corner $(x_i, y_i)$ to $(\hat{x}_i, \hat{y}_i)$, then we crop the aligned reference patch $\hat{P}_{Ho}^i$ from $I_{Ho}$. On the right, we compare the reselected patch $P_{Ls}^i$, unaligned Ref patch $P_{Ho}^i$ and the aligned Ref patch $\hat{P}_{Ho}^i$.

then incorporated into the cross-attention mechanism as an additive bias to the reference key features:

$$\text{Cross\_attn}_{opt} = \text{Softmax}\left(\frac{Q[K, K_{\text{ref}} + E_{Lo \to Ls}]^\top}{\sqrt{d}}\right)[V, V_{\text{ref}}], \tag{4}$$

where $E_{Lo \to Ls}$ denotes the motion embeddings derived from the dense pixel displacement field $O_{Lo \to Ls}$. The encoded relative motion acts as a spatial bias that facilitates the query to attend to aligned regions, thereby improving restoration under misaligned scenarios.

### 3.4.2 IMAGE SPACE MOTION ALIGNMENT

Live Photos typically have ultra-high resolutions (*i.e.*, $3072 \times 4096$ for the original key photo), which require patch-wise inference through a tiling strategy due to GPU memory limits. However, subject motion often causes pixel-level misalignment between patches of $I_{Ls}$ and $I_{Ho}$, leading to content inconsistency that undermines reference-guided restoration. To mitigate this issue, and as a complement to motion-guided attention, we propose a *Patch Correspondence Retrieval (PCR)* strategy to stabilize local matching and improve content alignment between the reselected patches from $I_{Ls}$ and the reference patches from $I_{Ho}$ during inference. Integrated into the tiling process and applied before the VAE encoder, it uses the estimated displacement field to locate the reference patch, ensuring that latent-space processing operates on spatially aligned inputs. Unlike pixel-wise warping, the proposed alignment is performed at the patch level rather than on individual pixels, which is consistent with the tiling-based inference pipeline and naturally preserves spatial consistency. As shown in Fig. 3, we first estimate the optical flow $O_{Ls \to Lo}$ from $I_{Ls}$ to $I_{Lo}$. For a patch $P_{Ls}^i \in \mathbb{R}^{p \times p \times 3}$ cropped from $I_{Ls}$ with the standard tiling strategy, we compute the average displacement vector within the patch as:

$$(\Delta x_i, \Delta y_i) = \left(\frac{1}{p^2}\sum_{j=1}^{p \times p} f_{x_i}^j, \frac{1}{p^2}\sum_{j=1}^{p \times p} f_{y_i}^j\right), \tag{5}$$

where $(f_{x_i}^j, f_{y_i}^j)$ represents the x- and y-axis components of the dense displacement field at pixel $j$, obtained from $O_{Ls \to Lo}$. The top-left corner $(\hat{x}_i, \hat{y}_i)$ of the aligned reference patch $\hat{P}_{Ho}^i$, cropped from $I_{Ho}$, is then computed by shifting the top-left corner $(x_i, y_i)$ of $P_{Ls}^i$:

$$(\hat{x}_i, \hat{y}_i) = (x_i + \Delta x_i, y_i + \Delta y_i). \tag{6}$$

The aligned reference patch $\hat{P}_{Ho}^i$ is then cropped from $I_{Ho}$ with the patch size $p$. As shown in Fig. 3, $\hat{P}_{Ho}^i$ achieves content consistency with $P_{Ls}^i$, compared to the unaligned reference patch that directly cropped from the same spatial location as $P_{Ls}^i$. With the tiling strategy, the corresponding optical flow patch is cropped at the same location as $\hat{P}_{Ho}^i$ to ensure spatial consistency of the network inputs.

## 4 EXPERIMENT

### 4.1 EXPERIMENTAL SETTINGS

**Training Datasets.** Since there are no existing datasets specifically for reselected key photo restoration in Live Photos, we construct the training set based on high-quality video data. For each sample, we select the HQ ground truth $I_{Hs}$ of the reselected frame and the original key photo $I_{Ho}$ with a clear temporal gap. In particular, we extract frames from the first 3,000 videos of the 2K-resolution DL3DV dataset (Ling et al., 2024), a large and multi-view video dataset covering over 10,000 scenes. The frame interval is set to 5, resulting in 50,400 image pairs at a resolution of $1024 \times 1024$. To further enrich the dataset with various motion patterns, we additionally collect 4K videos from the internet, obtaining 25,000 image pairs at the same resolution. To simulate degradations in Live Photo videos, we utilize the Real-ESRGAN (Wang et al., 2021) pipeline to obtain the LQ reselected frame $I_{Ls}$, with parameters specifically adjusted to better align the real-world degradations in mobile photography.

**Test Datasets.** For model evaluation, we construct a synthesized dataset, **SynLive260**, along with two real-world Live Photo datasets, **vivoLive144** and **iPhoneLive90**, collected from high-end consumer smartphones. VivoLive144 and iPhoneLive90 consist of 144 and 90 Live Photos captured with vivo X200 Pro and iPhone 15 Pro, respectively. In both datasets, the LQ reselected frames are manually extracted from the associated Live Photo video clips. These two datasets cover diverse dynamic scenes, including indoor and outdoor environments, street views, portraits, pets, and objects with motions from camera movement and subject dynamics. SynLive260 is built from 182 internet videos, covering various scenarios. After frame extraction, we obtain 260 image pairs of the HQ reselected frame $I_{Hs}$ and the original key photo $I_{Ho}$. The HQ reselected frame is then degraded using the Real-ESRGAN pipeline with a $\times 2$ downsampling factor to generate the corresponding LQ reselected frame $I_{Ls}$, simulating degradations in Live Photo videos.

**Evaluation Metrics.** On both the synthesized dataset and real-world Live Photo dataset, we adopt no-reference metrics for evaluation. We apply four metrics originally designed for SISR: NIQE (Zhang et al., 2015), MUSIQ (Ke et al., 2021), CLIPIQA (Wang et al., 2023a), and MANIQA (Yang et al., 2022). However, unlike traditional SR settings, our task provides access to the original high-quality key photo $I_{Ho}$ as a reference. This allows us to extend these metrics into a relative no-reference form by computing the normalized deviation between the restored reselected frame $\widetilde{I}_{Hs}$ and $I_{Ho}$:

$$\text{metric}_{re}(\widetilde{I}_{Hs}, I_{Ho}) = \frac{\left| \text{metric}(\widetilde{I}_{Hs}) - \text{metric}(I_{Ho}) \right|}{\text{metric}(I_{Ho})}, \tag{7}$$

where $\text{metric}(\cdot)$ denotes one of the above no-reference metrics. This formulation quantifies the quality gaps between the restored image $\widetilde{I}_{Hs}$ and a known HQ reference. To further assess perceptual alignment with $I_{Ho}$, we introduce two relative-reference metrics, CLIP-Q and DINO-Q, derived from CLIP-I and DINO from DreamBooth (Ruiz et al., 2023). Following CLIPIQA, we remove the positional embeddings from both models to support high-resolution inputs and eliminate resizing-induced distortions, which is crucial for faithful quality assessment. We also calculate full-reference metrics on the synthesized dataset: PSNR, SSIM (Wang et al., 2004), LPIPS (Zhang et al., 2018) and DISTS (Ding et al., 2020), where PSNR and SSIM are computed on y-channel in the YCbCr space. FID is also evaluated on SynLive260. Since it requires downsampling 4K images to a low resolution, we compute patch-wise FID by cropping images into $512 \times 512$ patches, which ensures that the evaluation resolution is consistent with recent restoration works (Dong et al., 2024).

**Implementation Details.** Both the ReferenceNet and the RestorationNet are initialized from the pre-trained weights of SD3-medium. We only train the MM-DiT backbones of both networks along with the Motion Encoder, while keeping the VAE encoder and decoder frozen. Training starts with an initial learning rate of $5e^{-5}$ and takes roughly 48h with 2 NVIDIA H20 GPUs and a batch size of 8. Inference is performed using a tiling strategy with a patch size of $1024 \times 1024$, which is kept identical to the training patch size for stable and fair evaluation.

### 4.2 COMPARISON WITH EXISTING METHODS

We compare LiveMoments with three categories of state-of-the-art methods: (a) RefISR methods, including TTSR (Yang et al., 2020), C2-Matching (Jiang et al., 2021), DATSR (Cao et al., 2022),

Table 1: Quantitative comparison with RefSR and SISR methods on real-world Live Photo datasets. The best results are highlighted in **bold**. Here, $NI_{re}$ denotes $NIQE_{re}$, $MU_{re}$ denotes $MUSIQ_{re}$, $CA_{re}$ denotes CLIPIQA and $MA_{re}$ denotes $MANIQA_{re}$.

| Method | vivoLive144 | | | | | | iPhoneLive90 | | | | | |
|---|---|---|---|---|---|---|---|---|---|---|---|---|
| | $NI_{re}\downarrow$ | $MU_{re}\downarrow$ | $CA_{re}\downarrow$ | $MA_{re}\downarrow$ | CLIP-Q↑ | DINO-Q↑ | $NI_{re}\downarrow$ | $MU_{re}\downarrow$ | $CA_{re}\downarrow$ | $MA_{re}\downarrow$ | CLIP-Q↑ | DINO-Q↑ |
| TTSR | 0.2050 | 0.3545 | 0.1928 | 0.1400 | 0.9626 | 0.8304 | 0.3069 | 0.2858 | 0.2593 | 0.1974 | 0.9505 | 0.8689 |
| C2-Matching | 0.2047 | 0.3512 | 0.1929 | 0.1390 | 0.9623 | 0.8298 | 0.3147 | 0.2800 | 0.2791 | 0.1978 | 0.9505 | 0.8685 |
| DATSR | 0.2212 | 0.3573 | 0.1951 | 0.1386 | 0.9621 | 0.8261 | 0.3324 | 0.2859 | 0.2853 | 0.1963 | 0.9500 | 0.8657 |
| MRefSR | 0.2276 | 0.3537 | 0.1916 | 0.1371 | 0.9616 | 0.8216 | 0.3408 | 0.2845 | 0.2900 | 0.1978 | 0.9499 | 0.8652 |
| CoSeR | 0.1953 | 0.1865 | 0.2752 | 0.1080 | 0.9658 | 0.9197 | 0.1774 | 0.2492 | 0.1564 | 0.0784 | 0.9608 | 0.8618 |
| ReFIR (SeeSR) | 0.3258 | 0.3042 | 0.8606 | 0.2731 | 0.9582 | 0.8156 | 0.2895 | 0.3605 | 0.5111 | 0.1429 | 0.9467 | 0.7319 |
| ReFIR (SUPIR) | 0.4665 | 0.2318 | 0.3571 | 0.1559 | 0.9201 | 0.8783 | 0.4336 | 0.2778 | 0.2224 | 0.0876 | 0.9190 | 0.8105 |
| RefVSR | 0.3798 | 0.3157 | 0.2142 | 0.1967 | 0.9609 | 0.8385 | 0.4226 | 0.2326 | 0.3016 | 0.2810 | 0.9472 | 0.8431 |
| ERVSR | 0.3314 | 0.4012 | 0.3457 | 0.1459 | 0.9597 | 0.8137 | 0.3635 | 0.3188 | 0.4569 | 0.2030 | 0.9448 | 0.8607 |
| StableSR | 0.3032 | 0.2833 | 0.7447 | 0.2250 | 0.9458 | 0.8491 | 0.2571 | 0.3566 | 0.3837 | 0.1253 | 0.9466 | 0.8174 |
| DiffBIR | 0.4228 | 0.3213 | 0.9547 | 0.3550 | 0.9023 | 0.7769 | 0.3477 | 0.3532 | 0.5397 | 0.1705 | 0.9145 | 0.7727 |
| SeeSR | 0.2767 | 0.2916 | 0.8168 | 0.2331 | 0.9606 | 0.8269 | 0.2957 | 0.3511 | 0.5174 | 0.1408 | 0.9445 | 0.7253 |
| SUPIR | 0.2703 | 0.2545 | 0.8275 | 0.1115 | 0.9407 | 0.8559 | 0.1805 | 0.3429 | 0.4924 | 0.0738 | 0.9422 | 0.7908 |
| OSEDiff | 0.2694 | 0.3191 | 0.8206 | 0.2444 | 0.9541 | 0.8536 | 0.2750 | 0.3832 | 0.4698 | 0.1260 | 0.9444 | 0.7525 |
| TSD-SR | 0.3359 | 0.3133 | 0.8519 | 0.2205 | 0.9476 | 0.8636 | 0.2963 | 0.3956 | 0.5142 | 0.1210 | 0.9477 | 0.7917 |
| LiveMoments | **0.0990** | **0.0893** | **0.0809** | **0.0556** | **0.9805** | **0.9629** | **0.0801** | **0.1230** | **0.1361** | **0.0543** | **0.9842** | **0.9466** |

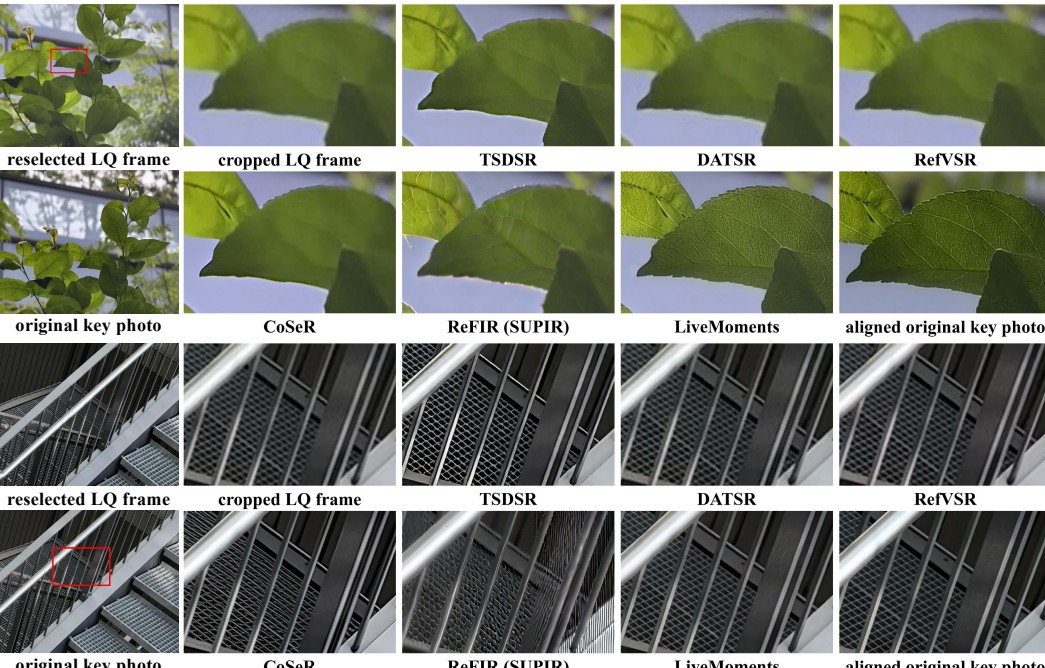

Figure 4: Visual comparison on the two real-world Live Photo datasets: vivoLive144 (top) and iPhoneLive90 (bottom). The aligned original key photo is cropped manually for better comparison.

and MRefSR (Zhang et al., 2023a), as well as diffusion-based methods CoSeR (Sun et al., 2024) and ReFIR (Yu et al., 2024) (based on SISR methods SeeSR and SUPIR). Designed for SISR, we replace the synthesized reference in CoSeR with the actual one. (b) RefVSR methods, including RefVSR (Lee et al., 2022) and ERVSR (Kim et al., 2023), for which we additionally supply full video inputs to match their model design. (c) Recent diffusion-based SISR methods, including multi-step methods (StableSR (Wang et al., 2024), DiffBIR (Lin et al., 2024), SeeSR (Wu et al., 2024b), SUPIR (Yu et al., 2024)) and single-step methods (OSEDiff (Wu et al., 2024a), TSD-SR (Dong et al., 2024)).

**Real-World Live Photo Datasets.** The quantitative results on the two Live Photo datasets are shown in Tab. 1, which can be found that our method achieves SOTA in both two datasets among all metrics. We note that commonly used no-reference metrics may be suboptimal for evaluating reference-guided restoration tasks, since they ignore the reference and often favor visually rich but inaccurate results. As illustrated in Fig. 5, a restored image with high quality scores but large deviation from the reference often contains visually implausible artifacts, whereas smaller deviations yield perceptually closer

Table 2: Quantitative comparison with RefSR and SISR methods on the synthetic dataset SynLive260. The best results are highlighted in **bold**. Here, $NI_{re}$ denotes $NIQE_{re}$, $MU_{re}$ denotes $MUSIQ_{re}$, $CA_{re}$ denotes CLIPIQA and $MA_{re}$ denotes $MANIQA_{re}$.

| Method | PSNR↑ | SSIM↑ | LPIPS↓ | DISTS↓ | FID↓ | $NI_{re}$↓ | $MU_{re}$↓ | $CA_{re}$↓ | $MA_{re}$↓ | CLIP-Q↑ | DINO-Q↑ |
|---|---|---|---|---|---|---|---|---|---|---|---|
| TTSR | 31.74 | 0.8778 | 0.2416 | 0.1259 | 18.53 | 0.4254 | 0.3885 | 0.1996 | 0.2209 | 0.9621 | 0.8373 |
| C2-Matching | **31.85** | 0.8782 | 0.2419 | 0.1250 | 18.51 | 0.4354 | 0.3873 | 0.2051 | 0.2199 | 0.9619 | 0.8391 |
| DATSR | 31.84 | 0.8783 | 0.2417 | 0.1251 | 18.50 | 0.4380 | 0.3876 | 0.2034 | 0.2186 | 0.9618 | 0.8388 |
| MRefSR | 31.84 | 0.8782 | 0.2420 | 0.1252 | 18.73 | 0.4342 | 0.3886 | 0.2058 | 0.2195 | 0.9612 | 0.8374 |
| CoSeR | 27.60 | 0.8136 | 0.2436 | 0.1135 | 13.92 | 0.2603 | 0.2548 | 0.6380 | 0.1028 | 0.9699 | 0.8924 |
| ReFIR (SeeSR) | 27.44 | 0.7942 | 0.2873 | 0.1530 | 28.26 | 0.3378 | 0.4028 | 1.2814 | 0.1837 | 0.9558 | 0.7934 |
| ReFIR (SUPIR) | 24.20 | 0.7024 | 0.3992 | 0.1779 | 33.95 | 0.4968 | 0.3273 | 0.7808 | 0.1491 | 0.9238 | 0.8295 |
| RefVSR | 26.28 | 0.8037 | 0.2937 | 0.1183 | 15.10 | 0.3713 | 0.3003 | 0.1964 | 0.2783 | 0.9631 | 0.8609 |
| ERVSR | 26.26 | 0.8016 | 0.3335 | 0.1306 | 20.22 | 0.4035 | 0.4021 | 0.2819 | 0.2129 | 0.9576 | 0.8327 |
| StableSR | 26.49 | 0.7619 | 0.3238 | 0.1491 | 23.77 | 0.3912 | 0.3951 | 1.1322 | 0.1573 | 0.9464 | 0.8386 |
| DiffBIR | 25.98 | 0.6927 | 0.4421 | 0.1958 | 36.39 | 0.3771 | 0.4199 | 1.3705 | 0.2260 | 0.9167 | 0.7824 |
| SeeSR | 27.25 | 0.7865 | 0.2981 | 0.1571 | 28.89 | 0.3602 | 0.3929 | 1.2805 | 0.1853 | 0.9538 | 0.7940 |
| SUPIR | 25.96 | 0.7309 | 0.3660 | 0.1762 | 31.53 | 0.2875 | 0.3741 | 1.3061 | 0.1083 | 0.9335 | 0.7869 |
| OSEDiff | 26.81 | 0.7882 | 0.2915 | 0.1412 | 27.17 | 0.3325 | 0.4132 | 1.1931 | 0.1669 | 0.9566 | 0.8220 |
| TSD-SR | 25.34 | 0.7379 | 0.3090 | 0.1625 | 24.03 | 0.4081 | 0.4595 | 1.3084 | 0.1641 | 0.9439 | 0.8208 |
| LiveMoments | 31.65 | **0.8990** | **0.0828** | **0.0365** | **4.00** | **0.0911** | **0.0720** | **0.1155** | **0.0495** | **0.9950** | **0.9740** |

Table 3: Ablation study of the network design on vivoLive144. All warp operations are applied on the original key photo, denoted as the reference image (Ref). The best results are highlighted in **bold**.

| Method | $NIQE_{re}$↓ | $CLIPIQA_{re}$↓ | $MANIQA_{re}$↓ | CLIP-Q↑ | DINO-Q↑ |
|---|---|---|---|---|---|
| RestorationNet | 0.1677 | 0.2348 | 0.1105 | 0.9690 | 0.9081 |
| RestorationNet + ReferenceNet | 0.1097 | 0.0823 | 0.0631 | 0.9792 | 0.9539 |
| RestorationNet + ReferenceNet + warp RefImage | 0.1034 | 0.0873 | 0.0573 | 0.9774 | 0.9480 |
| RestorationNet + ReferenceNet + warp RefLatent | 0.1130 | 0.0850 | 0.0622 | 0.9774 | 0.9437 |
| RestorationNet + ReferenceNet + warp RefKV | 0.1183 | 0.0853 | 0.0657 | 0.9776 | 0.9456 |
| LiveMoments (full model) | **0.0990** | **0.0809** | **0.0556** | **0.9805** | **0.9629** |

results. This reveals the limitation of no-reference metrics and highlights the effectiveness of the proposed relative no-reference metric. The visual results are provided in Fig. 4, where we compare representative methods with strong quantitative results. Our method effectively restores the reselected frame with details comparable to the original key photo without introducing artifacts, even when the input suffers from motion misalignment. More results are provided in the *supplementary material*.

**Synthetic Dataset.** The quantitative results on SynLive260 are shown in Tab. 2. It can be found that our method achieves the best performance among nearly all metrics. Although it shows lower PSNR, this mainly reflects the limitation of full-reference metrics in real-world restoration tasks, as discussed in (Yu et al., 2024). In contrast, the proposed task-specific metrics and qualitative comparisons (in the *supplementary material*) better capture the perceptual fidelity of the results.

### 4.3 ABLATION STUDY

**Effectiveness of the dual-branch architecture.** To validate the effectiveness of our dual-branch design, we first evaluate a baseline that uses only the RestorationNet without any reference guidance. As shown in the 1-st row of Tab. 3, performance drops across all metrics, highlighting the importance of introducing the ReferenceNet and the original key photo to enhance restoration quality.

**Effectiveness of the motion-guided attention.** We evaluate the effectiveness of our motion-guided attention by exploring different strategies for incorporating dense correspondence for motion alignment into the network. Specifically, we evaluate a model without latent-level alignment and several variants that inject motion information at different stages of the ReferenceNet, as shown in rows 2-5 in Tab. 3. All models are evaluated under our patch correspondence retrieval strategy to ensure spatial consistency during tiled inference on our vivoLive144 dataset. Among them, the proposed motion-guided attention outperforms other variants. This demonstrates the advantage of injecting alignment to guide the feature matching within the cross-attention mechanism, rather than relying on explicit warping. We provide the results on SynLive260 in the *supplementary material*.

**Effectiveness of the Patch Correspondence Retrieval (PCR).** We present challenging case with large motion and complex textures that hinder patch-level alignment, to validate the effectiveness of

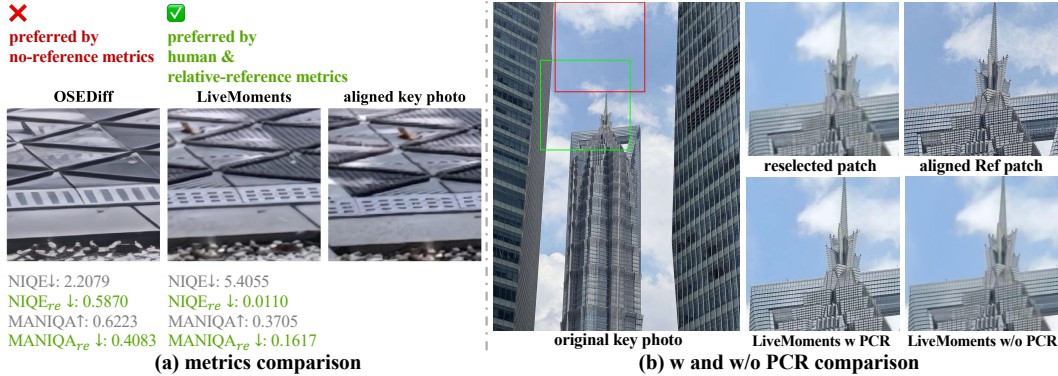

Figure 5: (a) Comparison between no-reference metrics, the proposed relative no-reference metrics, and human visual preference in evaluating the proposed task. The aligned key photo is cropped manually for better comparison. (b) Comparison between LiveMoments with (w) and without (w/o) the proposed Patch Correspondence Retrieval (PCR) strategy when processing 4K images. Patches are cropped for clearer visualization.

PCR. As shown in Fig. 5, our method establishes accurate correspondence between the two inputs, leading to more consistent and visually coherent restoration. These results highlight the robustness of our strategy under real-world scenes. We provide more visual results in the *supplementary material*.

## 5 CONCLUSION

We propose LiveMoments, a diffusion-based network designed for reselected key photo restoration in Live Photos. LiveMoments employs a dual-branch neural network with cross-attention between the branches to transfer detailed structural and textural information from the original high-quality key photo to the reselected low-quality frame. To address the challenge of large motion misalignment, we introduce a unified Motion Alignment module that facilitates alignment between the reselected and original key photos in both the latent and image spaces. Furthermore, we present a dedicated benchmark comprising both real and synthetic Live Photos, along with task-specific evaluation metrics for comprehensive comparison. Extensive experiments on three datasets demonstrate that LiveMoments significantly outperforms existing methods across both quantitative metrics and visual quality, particularly in challenging scenarios.

## ETHICS STATEMENT

Our work focuses on restoring reselected key photos in Live Photos for improving mobile photography. The synthetic dataset SynLive260 is built from publicly available video sources with no personal information. The real-world datasets (vivoLive144 and iPhoneLive90) were collected with the consent of participants. Before public release, all identifiable faces will be removed or anonymized to minimize privacy concerns. Our method is not intended for identity recognition or generation, and the released resources will be restricted to academic research in image restoration. Potential misuse is minimal given the planned anonymization and the academic scope of this work.

## REPRODUCIBILITY STATEMENT

We provide implementation details in Sec. 4.1 and Appendix B, covering both training and inference. Details of dataset preparation and evaluation metrics are also included in Sec. 4.1. We plan to release our code and datasets to enable reproducibility and encourage further research.

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

# A    THE USE OF LARGE LANGUAGE MODELS (LLMS)

We used large language models (LLMs) only to polish the writing and check grammar. They were not used to generate ideas, design methods, or influence experimental results.

# B    MORE IMPLEMENTATION DETAILS

## B.1    LOSS DETAILS

We define the forward process of the flow matching as,

$$z_t = \alpha(t)z_{Hs} + \beta(t)z_{Ls} + \sigma(t)\epsilon, \tag{8}$$

and the objective:

$$\mathcal{L}_\theta = \mathbb{E}_{t,z_t} ||G_\theta(z_t, t) - \frac{dz_t}{dt}||_2^2. \tag{9}$$

Specifically, we define the coefficients as,

$$\alpha(t) = \frac{(1-t)(0.2-t)}{0.2}, \quad \beta(t) = \frac{(1-t)t}{0.2}, \tag{10}$$

$$\sigma(t) = t, \qquad \text{with} \quad t \in [0, 0.2]. \tag{11}$$

It can be observed that $\alpha(t)$, $\beta(t)$ and $\sigma(t)$ are differentiable and sum to 1 for every $t$. Moreover, when $t = 0$, $z_t = z_{H_s}$, and when $t = 0.2$, $z_t = 0.8z_{L_s} + 0.2\epsilon$. This formulation adheres to the flow-matching framework, with the only deviation from the classical setup being the initial distribution: instead of pure Gaussian noise, we use a biased mixture defined as $z_{mix} = 0.8z_{L_s} + 0.2\epsilon$. The learning objective is to estimate a velocity field that transports $z_{mix} \rightarrow z_{H_s}$ within a narrow time window of length 0.2.

The core intuition is that low-quality (*i.e.*, $z_{Ls}$) and high-quality (*i.e.*, $z_{Hs}$) images share most of their low-frequency information. In other words, the primary difference between them lies in the high-frequency details. As a result, applying the full computational cost of denoising a completely random image—as is done in classical diffusion or flow-matching approaches—is largely inefficient.

Therefore, we apply only a 20% Gaussian noise perturbation, which preserves low-frequency information while introducing stochasticity to maintain a well-posed learning problem, *e.g.*, preventing the collapse of the generative diffusion prior. Furthermore, the ODE is integrated over a shortened time interval, $t \in [0, 0.2]$, significantly reducing the sampling path length. As a result, our inference process requires only 6 sampling steps, making it substantially faster than most diffusion-based image generation models. More importantly, due to the reduced stochasticity, the predictions become more deterministic—an advantageous property in image restoration scenarios where consistency is critical.

## B.2    TRAINING DETAILS

Following prior work, we perform cross-attention across all DiTs in Stable Diffusion 3 Medium for full feature fusion, involving both image and text embeddings. Since the reference image provides rich structural and textural cues, we use an empty text prompt during training and inference. We fine-tune the image branches in both the DiT-based ReferenceNet and RestorationNet, while keeping the text branch fixed to fully exploit the HQ guidance.

## B.3    COMPARISONS ON COMPUTATIONAL COSTS

LiveMoments processes a **1024×1024 patch** in **1.89 seconds** under FP16 inference on an H20 GPU (14.6 GB memory, 98.91 TFLOPs). For a 4K Live Photo (*i.e.*, $3072 \times 4096$), the total processing time is about **40 seconds**. To further clarify the computational cost, we report the parameter count, as well as the parameter-only VRAM and the inference-time peak VRAM for all components of LiveMoments in Tab. 4, measured when processing a 4K Live Photo.

We also provide a detailed comparison of parameter count, FLOPs, peak memory usage, and inference time between our method and the diffusion-based RefISR and SISR methods. All measurements were conducted on a single NVIDIA H20 GPU and run in mixed precision under a resolution of

Table 4: Comparisons on branch-wise parameter counts and VRAM, measured when processing a 4K Live Photo.

| Component | Parameter Counts (M) | Parameter-only VRAM (MB) | Inference-time Peak VRAM (MB) |
|---|---|---|---|
| VAE | 83.82 | 159.87 | 12368.33 |
| ReferenceNet | 2028.33 | 3868.73 | 10090.78 |
| RestorationNet | 2084.95 | 3976.73 | 10091.28 |
| RAFT | 5.26 | 10.03 | 13180.38 |
| Motion Encoder | 66.09 | 126.06 | 9940.00 |

Table 5: Comparisons with RefISR and SISR methods on parameter counts, FLOPs, peak memory usage and inference time at a resolution of $1024 \times 1024$ on a single NVIDIA H20 GPU.

| Type | Method | Parameter Counts (M) | TFLOPs | Peak Memory (GB) | Inference Time (s) |
|---|---|---|---|---|---|
| Time-Step Distilled Methods | OSEDiff | 1294.38 + 470.93 (DAPE) | 20.53 | 8.66 | 0.43 |
| | TSD-SR | 2207.33 | 21.91 | 8.60 | 0.24 |
| Non-Distilled Methods | CoSeR | 2655.52 | – | 32.74 | 48.83 |
| | ReFIR (SeeSR) | 2039.83 + 470.93 (DAPE) | 1560.70 | 17.83 | 27.62 |
| | ReFIR (SUPIR) | 4801.18 | 2672.70 | 59.82 | 31.67 |
| | StableSR | 1554.64 | 2236.31 | 34.53 | 164.67 |
| | DiffBIR | 1683.45 | 691.43 | 35.46 | 26.02 |
| | SeeSR | 2039.83 + 470.93 (DAPE) | 741.42 | 12.07 | 14.54 |
| | SUPIR | 4801.18 | 1200.65 | 54.25 | 16.98 |
| | **LiveMoments** | 4268.45 | 98.91 | 14.53 | 1.89 |

$1024 \times 1024$, except for CoSeR and DiffBIR, which do not support mixed precision inference at that resolution. As shown in Tab. 5, our LiveMoments achieves the lowest TFLOPs and the shortest inference time among the multi-step methods, while maintaining a competitive model size and peak memory usage to others. Although one-step diffusion methods such as OSEDiff and TSDSR benefit from distillation-based training and therefore exhibit faster inference and lower computational cost, LiveMoments achieves a trade-off between computational cost and restoration quality.

## C   USER STUDY

We compare LiveMoments with three representative RefSR methods: a traditional method (DATSR), a diffusion-based approach (ReFIR), and the strongest quantitative baseline (CoSeR). A total of 26 participants took part in the vote. They were asked to select the result that best matches the visual quality of the reference while preserving the content of the reselected LQ frame. As shown in Fig. 6, our method receives 69.62% approval rates, indicating its effectiveness.

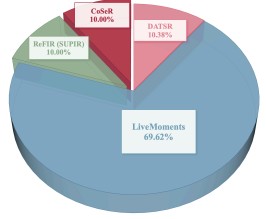

Figure 6: User study results.

## D   MORE ANALYSIS OF THE PROPOSED RELATIVE NO-REFERENCE METRICS

### D.1   CORRELATION BETWEEN RELATIVE NO-REFERENCE METRICS AND HUMAN PREFERENCES

Using the same three baselines as in the main user study (DATSR, ReFIR, and CoSeR), we further conducted a ranking-based experiment to analyze the correlation between the relative no-reference metrics and human preferences. We randomly sampled 15 images from vivoLive144 and 10 images from iPhoneLive90. A total of 15 participants are invited to rank the images based on perceptual similarity to the reference.

In Tab. 6, the Spearman and Pearson correlations show that all four metrics maintain positive, moderate alignment with human perception across devices and ISPs. These values are comparable to those reported in generic NR-IQA methods without task-specific training such as CLIPIQA (typically 0.36–0.74), despite our much smaller real-world dataset. This indicates that the proposed relative no-reference metrics remain meaningfully correlated with human perception.

Table 6: Spearman and Pearson correlations between relative no-reference metrics and human perception on real-world Live Photo datasets.

| Metric | vivoLive144 | | iPhoneLive90 | |
|---|---|---|---|---|
| | Spearman↑ | Pearson↑ | Spearman↑ | Pearson↑ |
| $NIQE_{re}$ | 0.493 | 0.518 | 0.604 | 0.560 |
| $MUSIQ_{re}$ | 0.585 | 0.543 | 0.414 | 0.440 |
| $CLIPIQA_{re}$ | 0.558 | 0.542 | 0.486 | 0.495 |
| $MANIQA_{re}$ | 0.535 | 0.548 | 0.370 | 0.413 |

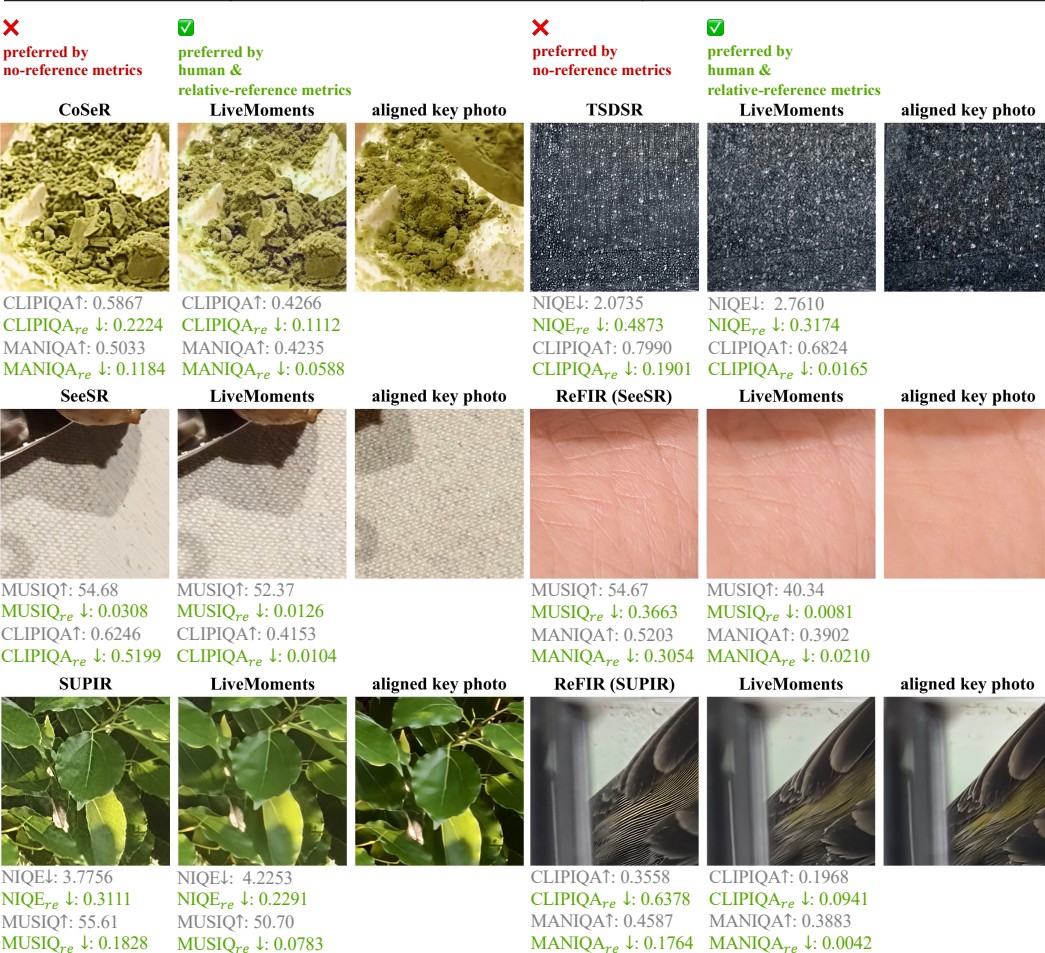

Figure 7: Comparison between no-reference metrics, the proposed relative no-reference metrics, and human visual preference in evaluating the key photo reselection task. The aligned key photo is cropped manually for better comparison.

## D.2 COMPARISON OF NO-REFERENCE METRICS WITH THE PROPOSED RELATIVE NO-REFERENCE METRICS

We present additional experiments in Fig. 7 to demonstrate that the widely-used no-reference metrics may have limitations in assessing image quality under the task of key photo reselection. It can be observed that these metrics prefer rich texture but far from real appearance results.

## E ADDITIONAL ABLATION RESULTS

### E.1 ABLATION STUDY ON SYNLIVE260 DATASET

We additionally provide ablation results on SynLive260 with ground-truth in Table 7. While our method does not achieve the absolute best scores in distortion-oriented metrics such as PSNR and

Table 7: Ablation study of the network design on SynLive260. All warp operations are applied on the original key photo, denoted as the reference image (Ref). The best results are highlighted in **bold**. Here, CA$_{re}$ denotes CLIPIQA and MA$_{re}$ denotes MANIQA$_{re}$.

| Method | PSNR↑ | SSIM↑ | LPIPS↓ | DISTS↓ | CA$_{re}$ ↓ | MA$_{re}$ ↓ | CLIP-Q↑ | DINO-Q↑ |
|---|---|---|---|---|---|---|---|---|
| RestorationNet | 30.42 | 0.8610 | 0.1685 | 0.0793 | 0.1721 | 0.0927 | 0.9841 | 0.9195 |
| RestorationNet + ReferenceNet | 31.47 | 0.8980 | 0.0837 | 0.0388 | 0.1246 | 0.0586 | 0.9949 | 0.9643 |
| RestorationNet + ReferenceNet + warp RefImage | 31.49 | 0.8927 | 0.0968 | 0.0475 | 0.1569 | 0.0546 | 0.9934 | 0.9556 |
| RestorationNet + ReferenceNet + warp RefLatent | **31.84** | **0.8986** | 0.0924 | 0.0441 | 0.1305 | 0.0618 | 0.9936 | 0.9566 |
| RestorationNet + ReferenceNet + warp RefKV | 31.82 | 0.8991 | 0.0895 | 0.0435 | 0.1306 | 0.0645 | 0.9940 | 0.9580 |
| LiveMoments (full model) | 31.65 | 0.8990 | **0.0828** | **0.0365** | **0.1155** | **0.0495** | **0.9950** | **0.9740** |

Table 8: Ablation study of different training degradation settings on vivoLive144. "Customized Live Photo degradation" is the setting used in the main experiments. Best results are highlighted in **bold**.

| Degradation Type | NIQE$_{re}$ ↓ | MUSIQ$_{re}$ ↓ | CLIPIQA$_{re}$ ↓ | MANIQA$_{re}$ ↓ | CLIP-Q↑ | DINO-Q↑ |
|---|---|---|---|---|---|---|
| Moderate-StableSR | 0.0925 | 0.1086 | 0.0883 | 0.0673 | 0.9786 | 0.9550 |
| StableSR | 0.0933 | 0.1165 | 0.1056 | 0.0734 | 0.9791 | 0.9581 |
| SeeSR | **0.0904** | 0.1030 | 0.0812 | 0.0647 | 0.9788 | 0.9505 |
| Customized Live Photo Degradation | 0.0990 | **0.0893** | **0.0809** | **0.0556** | **0.9805** | **0.9629** |

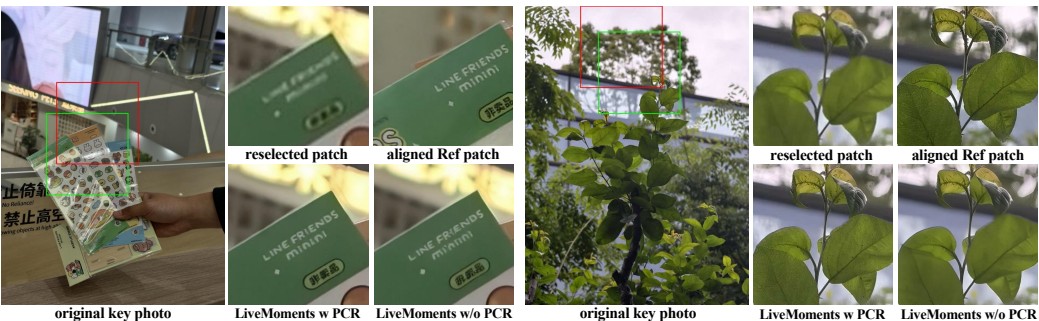

Figure 8: Comparison between LiveMoments with (w) and without (w/o) the proposed Patch Correspondence Retrieval (PCR) strategy when processing 4K images. Patches are cropped for clearer visualization. Please zoom in for better view.

SSIM, it consistently surpasses all baselines across perceptual and reference-based quality metrics (LPIPS, DISTS, CLIP-Q, DINO-Q, etc.). These results indicate that LiveMoments is more effective at recovering fine-grained structures and perceptual fidelity, aligning better with human perception. The consistent performance on both synthetic and real data further confirms the robustness and effectiveness of our design.

### E.2    MORE VISUAL EXAMPLES FOR PATCH CORRESPONDENCE RETRIEVAL (PCR)

To further validate the effectiveness of our Patch Correspondence Retrieval (PCR) strategy, we provide additional challenging cases in Fig. 8. Even under large motion and complex textures, PCR corrects spatial offsets between degraded and reference patches, enabling accurate transfer of fine details such as text edges and leaf veins. These results confirm that PCR enhances both restoration fidelity and fine-grained detail recovery in high-resolution scenarios.

### E.3    ANALYSIS ON DEGRADATION SETTINGS

We conducted analyses to explore how different synthetic degradation settings affect real-world performance. Specifically, we trained LiveMoments under three representative degradation settings: (1) SeeSR degradation, (2) StableSR degradation, and (3) a lighter moderate-StableSR variant constructed by reducing the blur and noise levels. The quantitative results on vivoLive144 are reported in Tab. 8. Among the three settings, our customized Live Photo degradation achieves the best results across most metrics, indicating that it better reflects real-world degradation characteristics. At the same time, LiveMoments maintains stable performance under all three degradation variants, demonstrating its robustness to different synthetic degradation.

We also provides the visualizations of degraded images in Fig. 9. For each setting, the synthetic degradation is applied to the original key photo from our real-world Live Photo dataset and compared

Table 9: Ablation study of robustness analysis under flow noise injection and flow estimator replacement on vivoLive144. Best results are highlighted in **bold**.

| Method | NIQE$_{re}\downarrow$ | MUSIQ$_{re}\downarrow$ | CLIPIQA$_{re}\downarrow$ | MANIQA$_{re}\downarrow$ | CLIP-Q$\uparrow$ | DINO-Q$\uparrow$ |
|---|---|---|---|---|---|---|
| LiveMoments + 10% noise | 0.0999 | 0.0921 | 0.0808 | 0.0561 | 0.9802 | 0.9622 |
| LiveMoments + 20% noise | 0.1035 | 0.0935 | **0.0802** | 0.0551 | 0.9799 | 0.9613 |
| LiveMoments + 40% noise | 0.1082 | 0.0993 | 0.0811 | **0.0546** | 0.9792 | 0.9591 |
| LiveMoments + 80% noise | 0.1108 | 0.1072 | 0.0845 | 0.0548 | 0.9785 | 0.9555 |
| LiveMoments + 100% noise | 0.1116 | 0.1099 | 0.0862 | 0.0566 | 0.9782 | 0.9540 |
| LiveMoments + 200% noise | 0.1159 | 0.1222 | 0.0915 | 0.0578 | 0.9771 | 0.9497 |
| LiveMoments (SPyNet) | 0.1006 | 0.0907 | 0.0809 | 0.0561 | 0.9804 | **0.9629** |
| LiveMoments (LiteFlowNet) | 0.1031 | 0.0946 | 0.0808 | 0.0544 | 0.9794 | 0.9595 |
| LiveMoments | **0.0990** | **0.0893** | 0.0809 | 0.0556 | **0.9805** | **0.9629** |

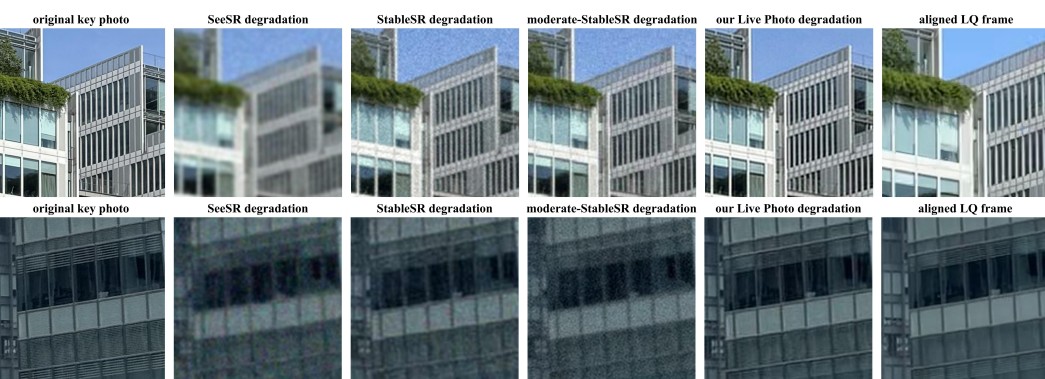

Figure 9: Comparison between different degradation settings. The aligned LQ frame is cropped manually for better comparison. Please zoom in for better view.

with the corresponding reselected LQ frame, allowing us to assess how well the synthetic degradations replicate real degradation patterns. The results show that existing settings tend to apply stronger degradations than those observed in real Live Photos, while our customized Live Photo degradations more faithfully reflect real-world characteristics.

### E.4    ROBUSTNESS ANALYSIS OF INACCURATE MOTION ALIGNMENT

We analyze the behavior and robustness of LiveMoments when the estimated optical flow is inaccurate and motion alignment becomes unreliable. To quantify its sensitivity to flow errors, we conduct two analyses on vivoLive144: (1) perturbing the RAFT flow with Gaussian noise, and (2) replacing RAFT with alternative flow estimators. The results is provided in Tab. 9.

**Noise Injection.** We add Gaussian noise with different magnitudes (10%–200% of the original flow magnitude) to the RAFT outputs. Although the perturbed flows become increasingly distorted, the reconstruction quality of LiveMoments degrades only slightly under moderate perturbations, indicating low sensitivity to flow inaccuracies.

**Flow Estimator Replacement.** We substitute RAFT with SPyNet and LiteFlowNet, and the results remain close to the original RAFT-based LiveMoments, suggesting that the model does not rely on a specific flow estimator and generalizes across different motion estimation methods.

In addition, we provide visualizations to further analyze the impact of flow errors, including the reselected LQ frame, the original key photo with optical flow, and the warped original key photo, together with cropped regions that illustrate local refinement quality.

**Robust cases.** As shown in Fig. 10, LiveMoments remains stable even when RAFT produces low-confidence or divergent flow vectors due to large motion or occlusion. Despite inaccurate motion guidance, the model does not introduce hallucinated textures and still produces visually coherent results, demonstrating its robustness to moderate flow errors.

**Failure cases.** In Fig. 11, we provide failure-case visualizations, including examples with large motion, severe occlusion, and low-texture regions that lead to inaccurate flow. These examples reveal the boundary conditions where motion alignment errors exceed the model's ability to compensate.

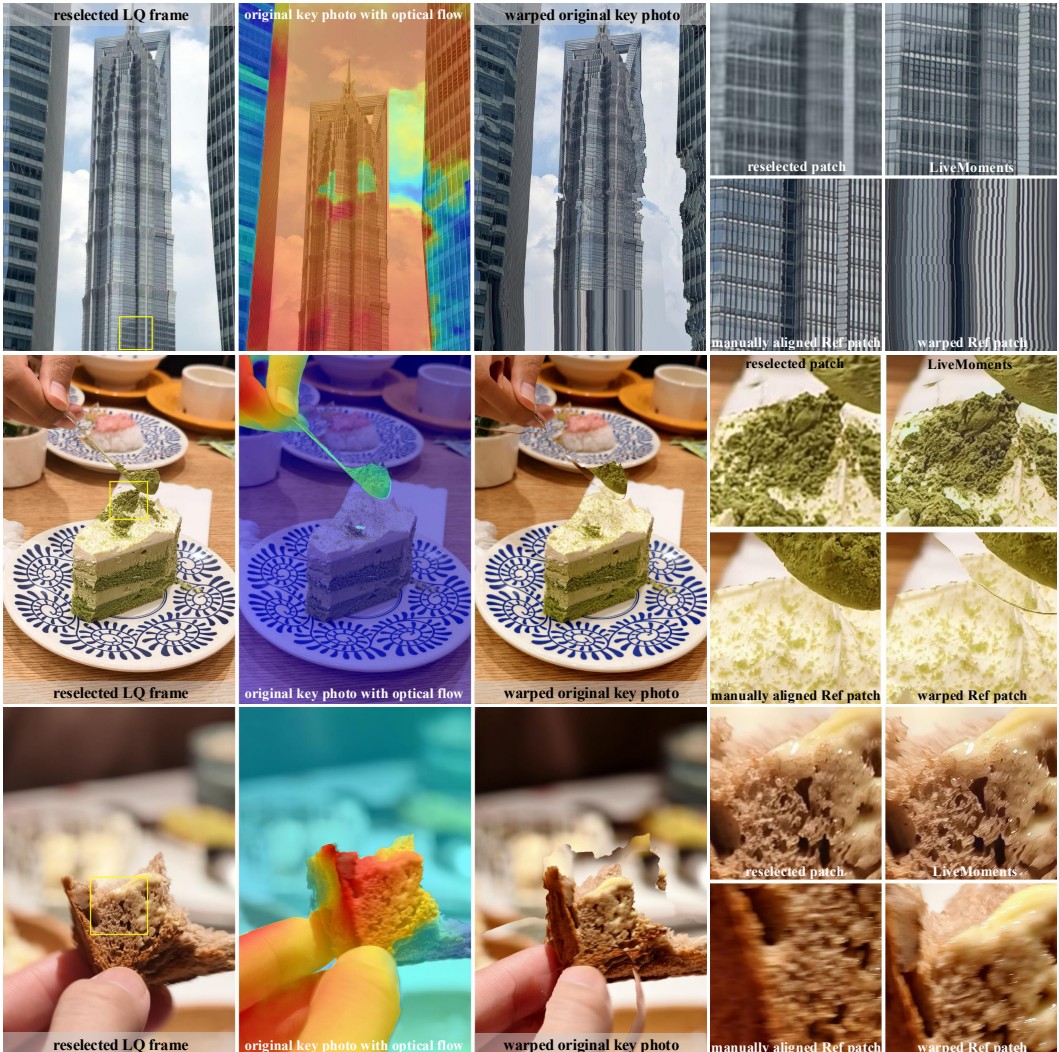

Figure 10: Robust cases with inaccurate motion alignment. The aligned original key photo is cropped manually for better comparison. Please zoom in for better view.

## F  MORE VISUAL RESULTS

In Figs. 12, 13, 14 and 16, we present additional visual comparisons on the real world Live Photo dataset **vivoLive144**, including cases under high-resolution scenes. We compare visual results on vivoLive144 from other RefISR, RefVSR and SISR baselines in Fig. 15. In Figs. 17 and 18, we present additional visual comparisons and high-resolution results on **iPhoneLive90**. The visual results of iPhoneLive90 from other RefISR, RefVSR and SISR baselines are provided in Fig. 19. In Figs. 20 and 21, we present additional visual comparisons and high-resolution results on **SynLive260**. These examples demonstrate the robustness of LiveMoments in addressing the key photo reselection task across scenarios.

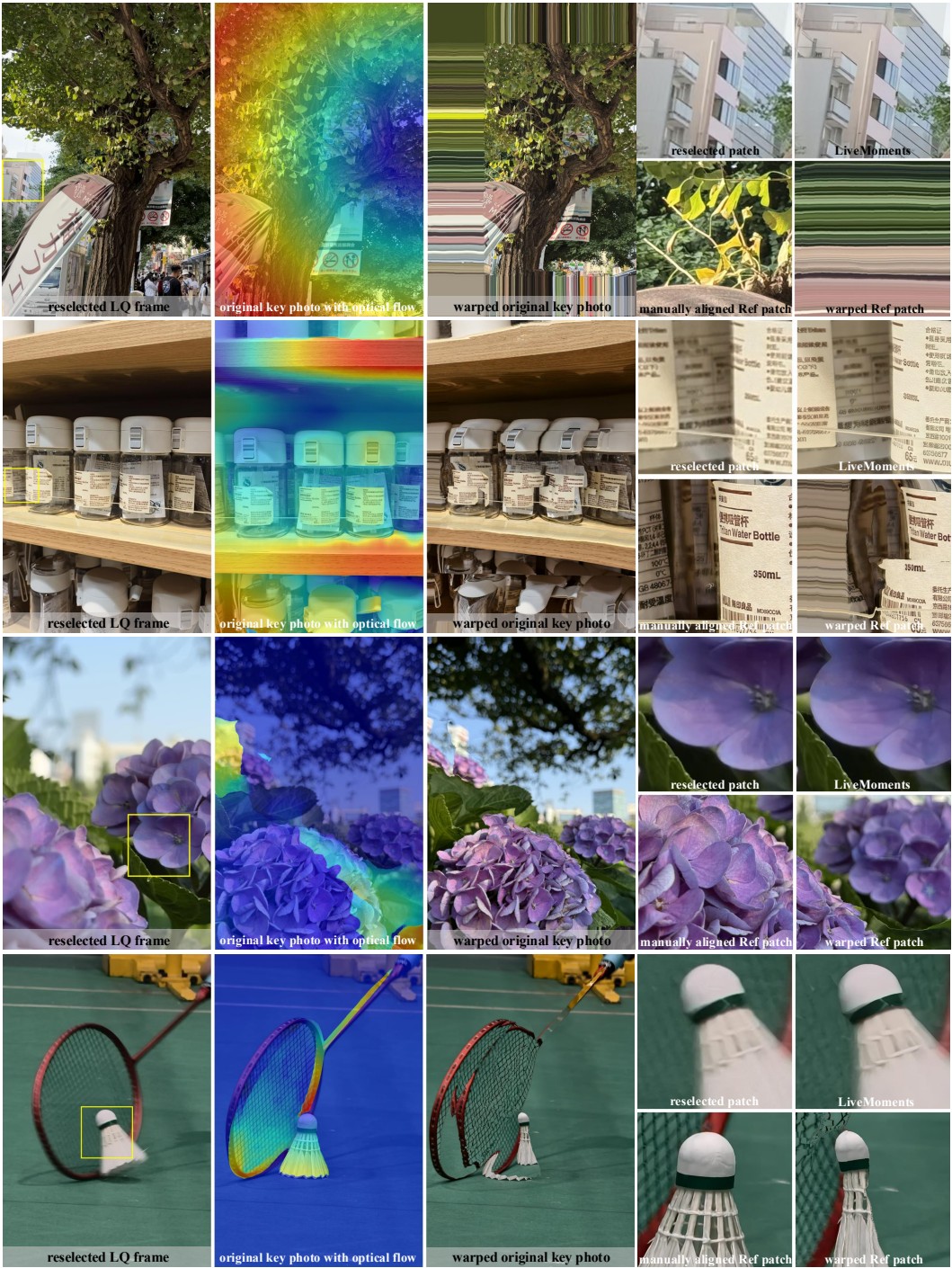

Figure 11: Failure cases with inaccurate motion alignment. The aligned original key photo is cropped manually for better comparison. Please zoom in for better view.

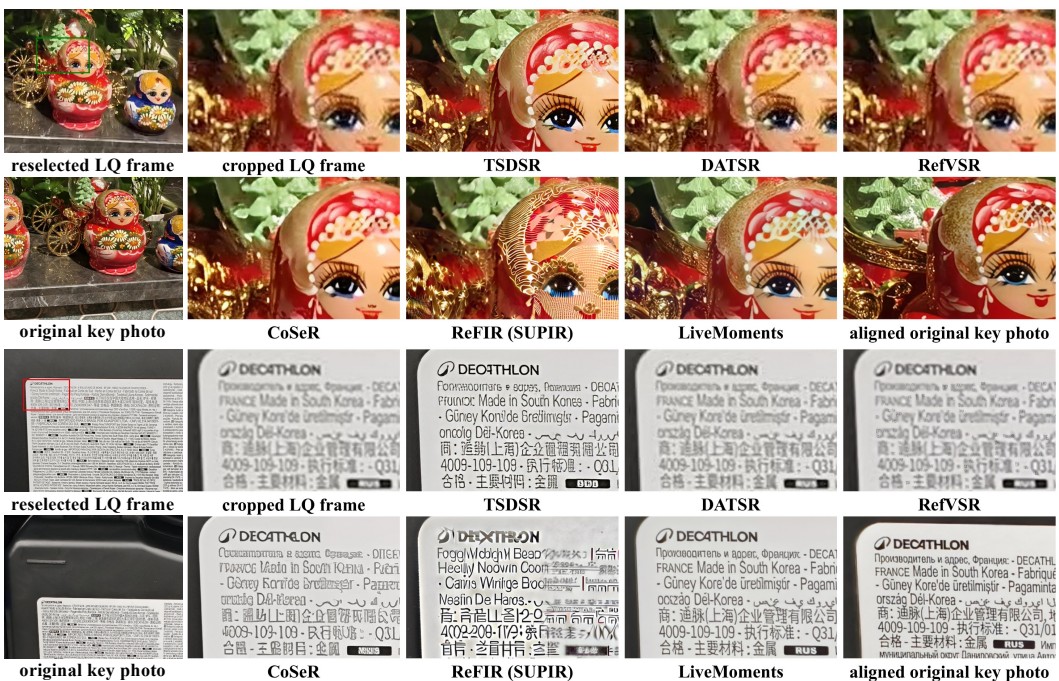

Figure 12: More visual comparisons of RefISR, RefVSR and SISR methods on vivoLive144 dataset. The aligned original key photo is cropped manually for comparison. Please zoom in for a better view.

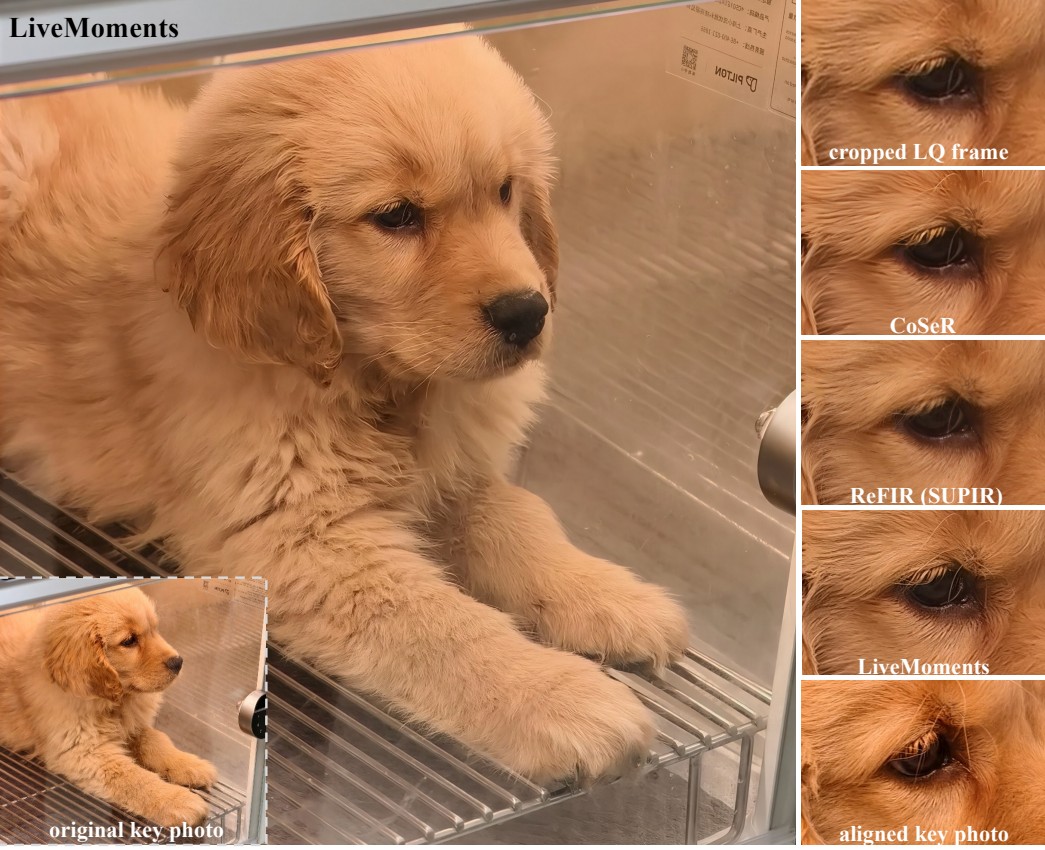

Figure 13: High-resolution visual comparisons of diffusion-based RefISR methods on vivoLive144 dataset. The aligned original key photo is cropped manually for better comparison.

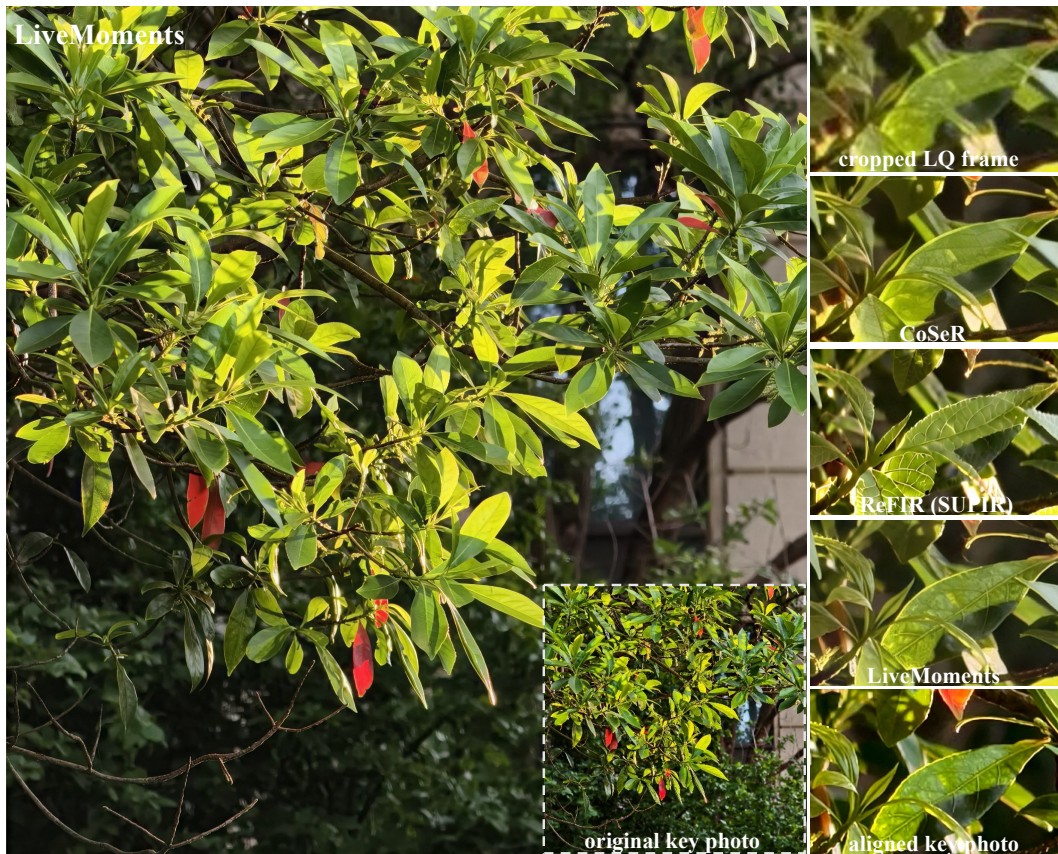

Figure 14: High-resolution visual comparisons of diffusion-based RefISR methods on vivoLive144 dataset. The aligned original key photo is cropped manually for better comparison.

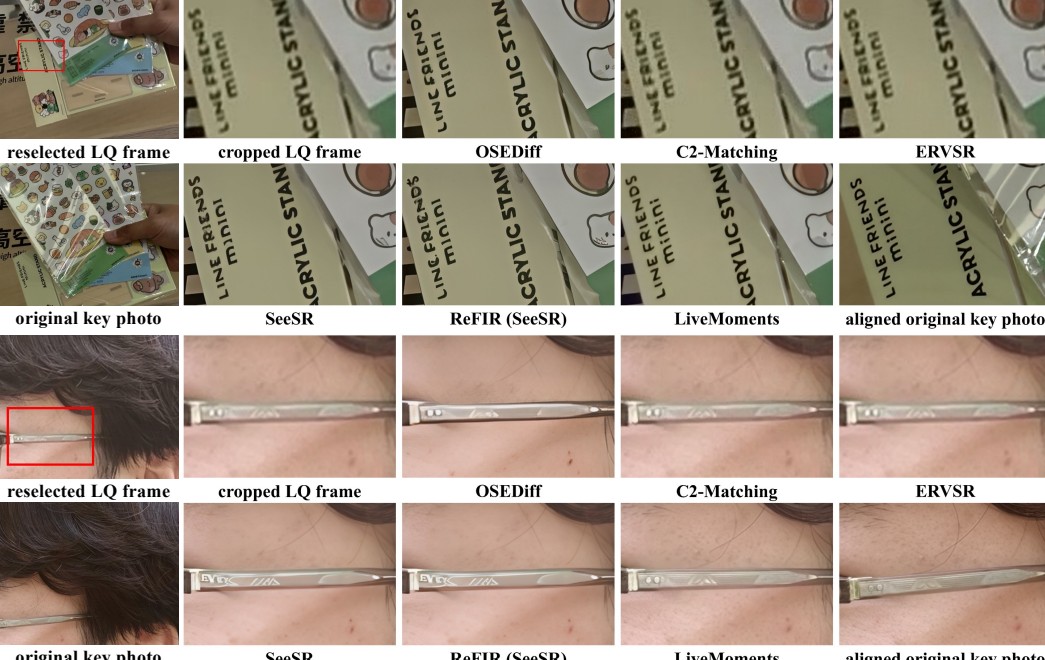

Figure 15: More visual comparisons of other RefISR, RefVSR and SISR methods on vivoLive144 dataset. The aligned original key photo is cropped manually for better comparison. Please zoom in for a better view.

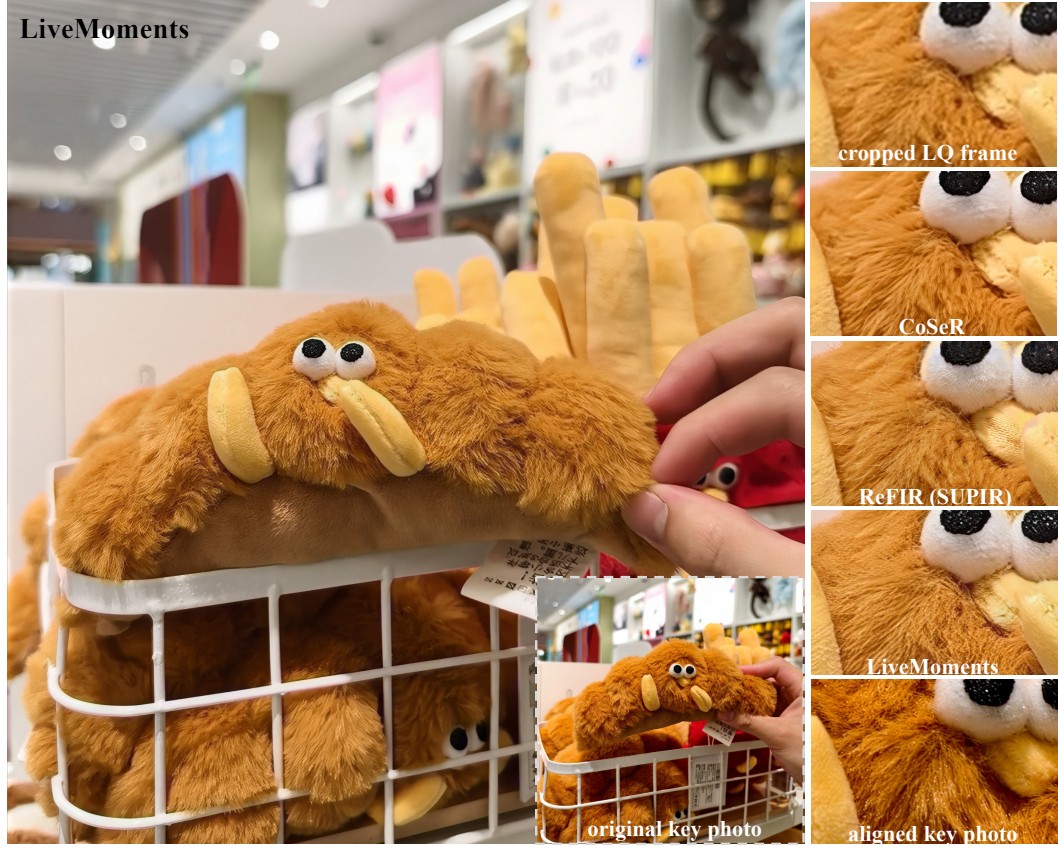

Figure 16: High-resolution visual comparisons of diffusion-based RefISR methods on vivoLive144 dataset. The aligned original key photo is cropped manually for better comparison.

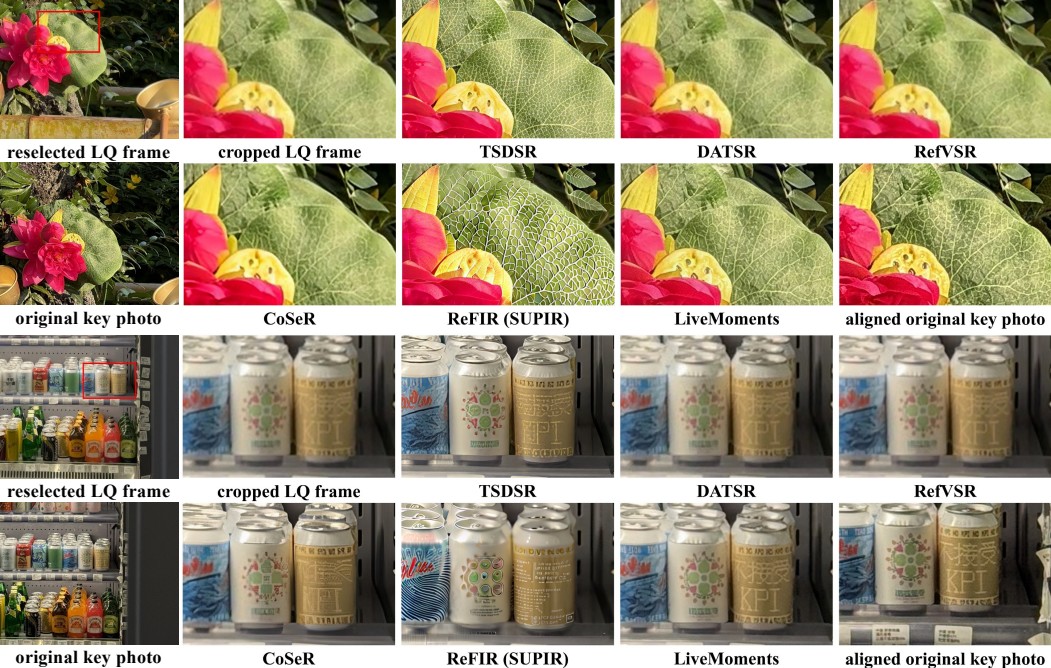

Figure 17: More visual comparisons of RefISR, RefVSR and SISR methods on iPhoneLive90 dataset. The aligned original key photo is cropped manually for comparison. Please zoom in for a better view.

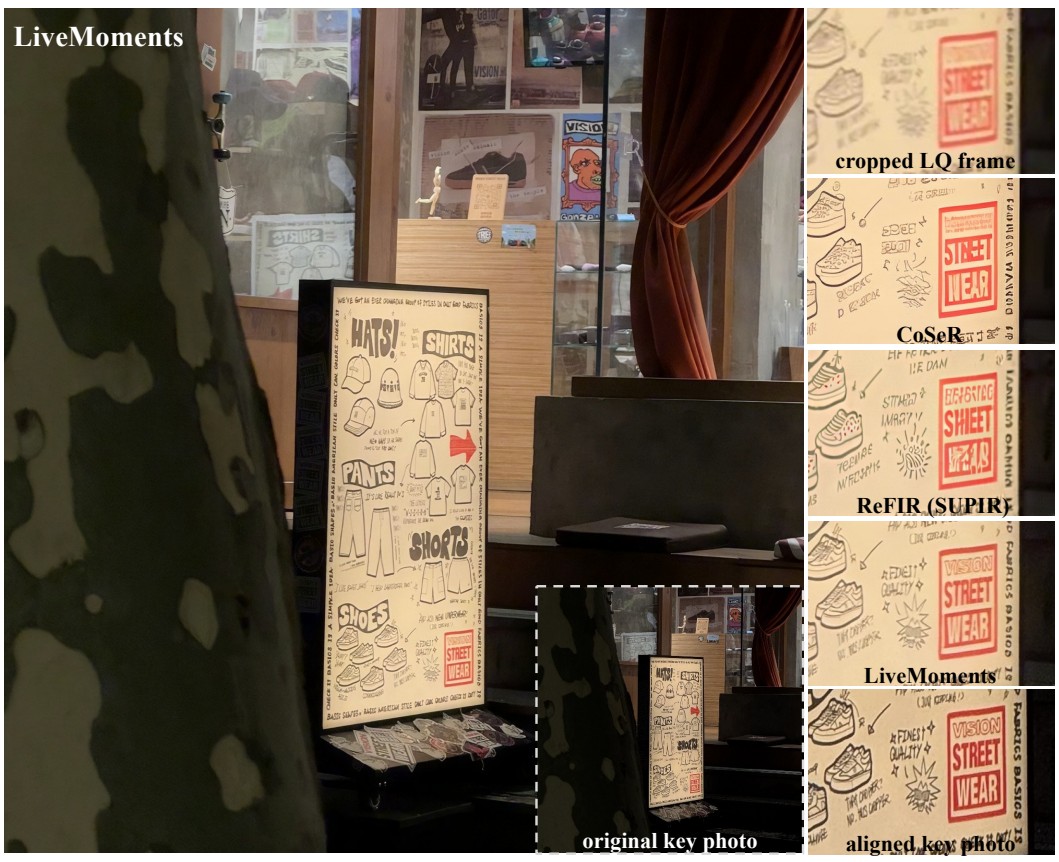

Figure 18: High-resolution visual comparisons of diffusion-based RefISR methods on iPhoneLive90 dataset. The aligned original key photo is cropped manually for better comparison.

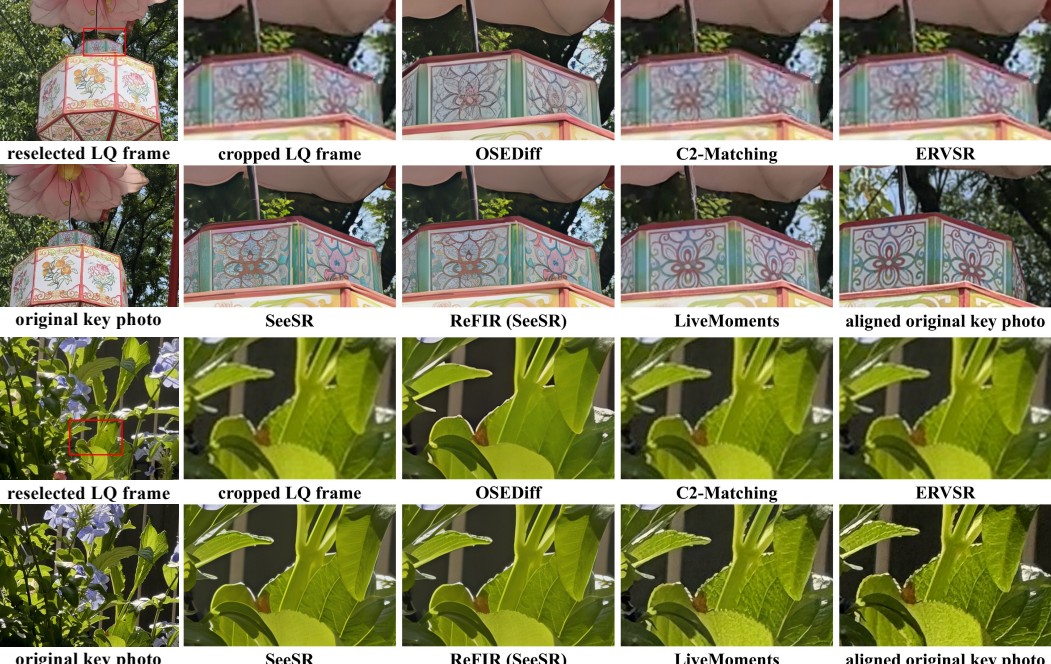

Figure 19: More visual comparisons of other RefISR, RefVSR and SISR methods on iPhoneLive90 dataset. The aligned original key photo is cropped manually for better comparison. Please zoom in for a better view.

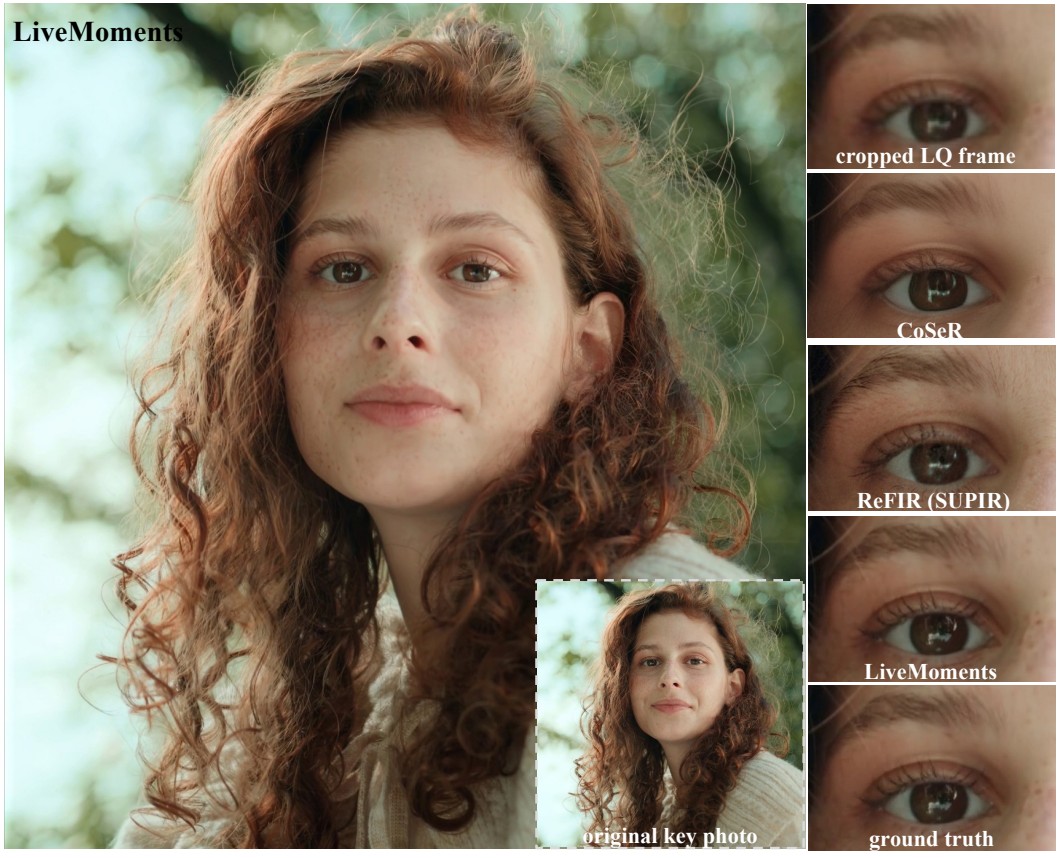

Figure 20: High-resolution visual comparisons of diffusion-based RefISR methods on SynLive260 dataset.

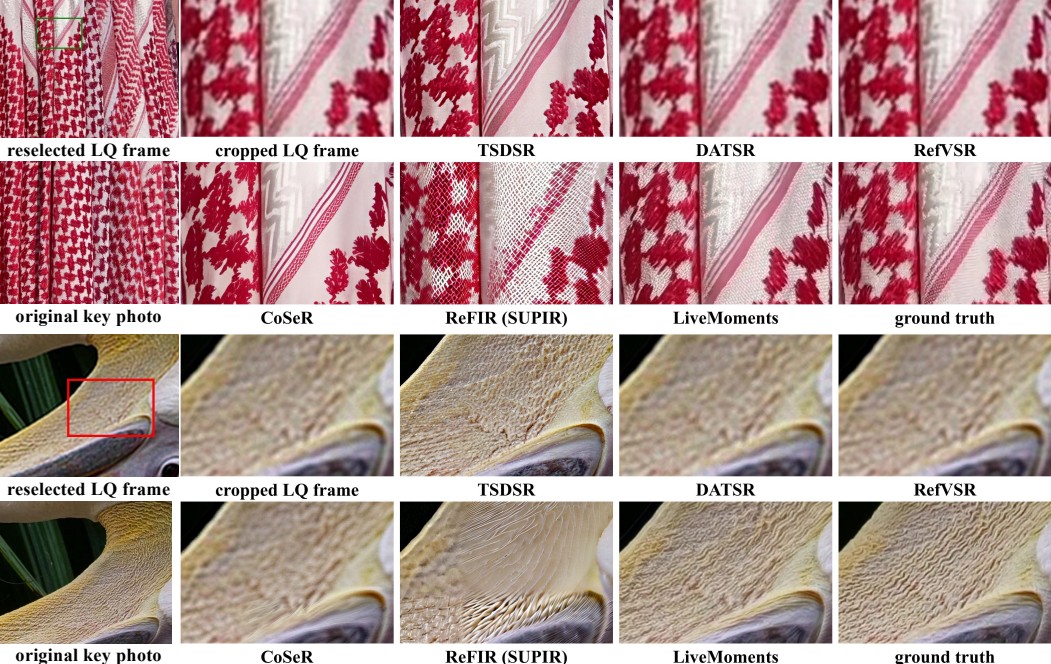

Figure 21: More visual comparisons of RefISR, RefVSR and SISR methods on SynLive260 dataset. Please zoom in for a better view.

