# OpenReview forum: "LiveMoments: Reselected Key Photo Restoration in Live Photos via Reference-guided Diffusion"
_ICLR.cc/2026/Conference — ICLR 2026 Poster_

### Official Review · Reviewer_HHsy · 2025-10-26

**Soundness:** 3
**Presentation:** 2
**Contribution:** 2
**Rating:** 4
**Confidence:** 3

**Summary:**

The paper presents LiveMoments, a method for selecting and restoring a new low-quality (LQ) key photo from a short clip surrounding some key high-quality (HQ) photo. To this end, the authors build a model based on latent flow models and learnable networks for the HQ key image, the LQ candidate, and the motion between the two frames modeled as optical flow. The authors also propose to perform image space motion alignment based on image patches. The authors train the model using open source high-quality data and introduce three benchmarks for evaluation, a synthetic one and two real-world Live Photo datasets.

**Strengths:**

* The paper introduces a novel task, Reselected Key Photo Restoration in Live Photos, and builds a model specifically designed for solving this task.
* The paper introduces novel benchmarks for this task.
* LiveMoments substantially outperforms competing methods from several related domains on the evaluated metrics.
* The paper is written clearly, although some parts are not clear, as will be detailed next.

**Weaknesses:**

* Although I appreciate the effort invested in this paper, I am not sure that, in terms of the scope and potential impact, it fits ICLR. The paper addresses a task that, in my opinion, is very nuanced, and it is challenging to see how to generalize the method to other tasks, setups, and types of image degradations. Could the authors please clarify the possible future directions for extending the proposed method and its other domains of applicability?
* To me, section 3.4.2 is not clear enough. For instance,
  - Is the space motion alignment done for each pixel? If so, then how is consistency preserved across pixels?
  - Is it applied to the output of the model or to the input?
  - How is the patch size selected? Perhaps an ablation about that can clarify it.
* Code is missing from the submission to self-evaluate the method quality and better understand the implementation.
* Minor:
  - Some information is missing about the vivoLive144 and iPhoneLive90 datasets. For instance, what types of scenes were used to construct these datasets?
  - Missing explanation for what is $f(x)$ in line 194 and $Q$ in line 232.
  - Why wasn't FID evaluated on the synthetic dataset?

**Questions:**

.

---

> ### Author Response · Authors · 2025-11-24
> **Response to Reviewer HHsy (1/2)**
>
> We sincerely thank you for the valuable feedback and constructive suggestions. In the revised manuscript, the corresponding revisions are highlighted in **purple**.
>
> ### **Potential extensions and broader applicability (Weakness 1)**
> Thank you for raising this important point. We would like to clarify that although our work is presented within the Live Photo setting, **the proposed framework is not limited to this specific scenario and is generally applicable beyond the LivePhoto setting**.
>
> **1. Applicability beyond the LivePhoto task.**
>
> LiveMoments is designed to restore a degraded target frame guided by a high-quality reference image that has temporal consistency but spatially misaligned. This setting appears in many real-world scenarios, such as adjusting color or tone of a reselected frame with stable reference, enhancing a video keyframe with information from a nearby sharp frame, or refining multi-capture mobile photography pipelines when some frames contain more details. In these cases, the key components in LiveMoments, such as ReferenceNet, RestorationNet and Motion Alignment module, can be adapted, showing that our method is not tied to the specific task.
>
> **2. Extensibility to multi-frame and multi-reference setups.**
>
> Although we focus on the single-reference setting to establish the new task, the model architecture does not inherently restrict extensions to multi-frame or multi-reference inputs. Multi-reference inputs can be incorporated by extending ReferenceNet to aggregate multiple features, and multi-frame setups can be supported by temporal-attention layers or KV caching. These modifications present natural future extensions.
>
> **3. Flexibility across different degradation types.**
>
> Extending LiveMoments to other types of image degradation is also straightforward. Since our framework learns a reference-guided transformation from degraded images to clean ones, it is not tied to any specific degradation type. Adapting to other types of degradation mainly requires changing the degradation used to construct the training set, such as applying color distortions, exposure inconsistencies, or stronger detail loss. The strong generative priors in diffusion models enables LiveMoments to generalize well under various degradation settings.
>
> We appreciate the reviewer's insightful feedback and hope that this clarification helps better reflect the scope and potential impact of our contribution.
>
> ### **Clarification on pixel-level motion alignment in Section 3.4.2 (Weakness 2)**
> Thank you for pointing out these unclear aspects. We apologize for the unclear explanation in the current draft and have revised Section 3.4.2 to clarify the proposed *Patch Correspondence Retrieval (PCR)* strategy in the revised paper. The details are explained below.
>
> 1. **Is the space motion alignment done for each pixel? If so, then how is consistency preserved across pixels?**: The term 'pixel-level alignment' refers to alignment performed in the pixel space (image domain) rather than on individual pixels, in contrast to the latent-space alignment described in the earlier section. During inference on 4K Live Photos, the diffusion model operates in a patch-wise manner. Therefore, our **PCR strategy is applied per patch**, using the estimated displacement field to locate the corresponding reference patch. Each patch is treated as a coherent unit, ensuring spatial consistency within the patch, while C²-blending is applied for smooth transitions across neighboring patches.
>
> 2. **Is alignment applied to the input or the output?**: The **alignment is applied to the input patches** before they are encoded by the VAE encoder during inference. After the VAE decoder, the decoded patches are stitched back using C²-blending.
>
> 3. **How is the patch size selected?**: The patch size is kept **identical to the training patch size** in our method. Using a different patch size at inference would change the receptive field and contextual range, introducing a train-test mismatch that may degrade stabiliy.  Following standard practice in high-resolution diffusion-based restoration [1], we adopt the same patch size for training and inference.
>
> ### **Code availability (Weakness 3)**
> Thank you for the suggestion. We have included an anonymized version of **the inference code and the core model architecture** in the updated supplementary materials. These files provide the full inference pipeline and the key components of our method.
>
> [1] Scaling Up to Excellence: Practicing Model Scaling for Photo-Realistic Image Restoration In the Wild. (Yu et al. CVPR 2024)

---

> > ### Author Response · Authors · 2025-11-24
> > **Response to Reviewer HHsy (2/2)**
> >
> > ### **Minor clarifications (Weakness 4)**
> > Thank you for the helpful comments. We would like to address the minor points below and have included these details in the revised manuscript.
> >
> > **1. Details of vivoLive144 and iPhoneLive90:** Both datasets cover a wide range of dynamic real-world scenarios, including indoor environments, outdoor natural scenes, street views, portraits, pets, and everyday objects. The dynamic nature of these scenes primarily comes from natural camera motions (e.g., displacement or field-of-view changes), subject motions or pose variations, and interactions with pets or moving objects, which together reflects realistic usage conditions across different devices.
> >
> > **2. Clarifications of $f(x)$ and $Q$:** In **equation at line 194**, $f(x)$ denotes neural network in the Rectified Flow framework which parameterizes the velocity field responsible for transforming samples from the data distribution $x_0 \sim p_0 $ to the noise $\epsilon \sim p_1$, via the forward process $x_t = (1-t)x_0 + t\epsilon$ as defined in the Rectified Flow literature [2][3]. In our work, this concept corresponds to the model $G_\theta$, which directly learns the velocity field between the source and target distributions. In **equation (3) at line 232**, Q refers to the query from the attention layer of the RestorationNet.
> >
> > **3. FID on Synthetic Data:** Following your suggestion, we **evaluate FID on our synthetic dataset SynLive260**.  Since computing FID requires downsampling 4K outputs to a low resolution (e.g., $299\times299$), which removes most high-frequency details, we instead crop each image into $512\times512$ patches and compute patch-wise FID. This makes our FID evaluation resolution consistent with that used in previous restoration works [4]. The results are provided in `Table 1`, where LiveMoments achieves the best FID performance.
> >
> > **Table 1: FID comparison with RefSR and SISR methods on the synthetic dataset SynLive260**
> > | Method                 |  FID $\downarrow$| Method               |  FID $\downarrow$ |
> > |------------------------|-------:|----------------------|-------:|
> > | TTSR                   | 18.53  | C2-Matching          | 18.51  |
> > | DATSR                  | 18.50  | MRefSR               | 18.73  |
> > | CoSeR                  | 13.92  |                      |        |
> > | ReFIR (SeeSR)          | 28.26  | ReFIR (SUPIR)        | 33.95  |
> > | RefVSR                 | 15.10  | ERVSR                | 20.22  |
> > | StableSR               | 23.77  | DiffBIR              | 36.39  |
> > | SeeSR                  | 28.89  | SUPIR                | 31.53  |
> > | OSEDiff                | 27.17  | TSDSR                | 24.03  |
> > | **LiveMoments (Ours)** |  4.00  |                      |        |
> >
> > [2] Flow Straight and Fast: Learning to Generate and Transfer Data with Rectified Flow. (Liu et al., ICLR 2023)
> >
> > [3] Scaling Rectified Flow Transformers for High-Resolution Image Synthesis. (Esser et al., ICML 2024)
> >
> > [4] TSD-SR: One-Step Diffusion with Target Score Distillation for Real-World Image Super-Resolution. (Dong et al. CVPR 2025)

---

> > > ### Comment · Reviewer_HHsy · 2025-11-27
> > >
> > > I thank the authors for the comments, clarifications, and the additional results. Following these additions I raise my score to 6.

---

### Official Review · Reviewer_cYvV · 2025-10-31

**Soundness:** 4
**Presentation:** 4
**Contribution:** 4
**Rating:** 8
**Confidence:** 4

**Summary:**

The paper introduces LiveMoments for reselected key photo restoration in Live Photos. It adopts a dual branch diffusion architecture with a ReferenceNet and a RestorationNet, and adds a unified Motion Alignment module that injects flow guided priors at latent and image levels. The authors build three benchmarks and propose relative no reference metrics tailored to the task. Experiments on synthetic and real Live Photo datasets demonstrate consistent gains over RefISR, RefVSR, and diffusion based SISR baselines.

**Strengths:**

- Clearly defined and practical new task specific to Live Photos with well motivated formulation
- Elegant dual branch diffusion design with cross attention that effectively transfers fine details from the original key photo
- Unified motion alignment at latent and image levels addresses large temporal offsets in real captures
- Benchmarks and task specific metrics are thoughtfully constructed and results are strong and consistent

**Weaknesses:**

- Efficiency comparison is limited  runtime and memory are reported for the proposed method only  a fair cross method efficiency table would strengthen the claim

**Questions:**

1. How sensitive is the method to errors in optical flow  can the authors include a noise injection or alternative flow estimator study and show qualitative failure cases where misalignment occurs
2. Can the authors provide a runtime and memory comparison to representative RefISR and diffusion restoration baselines under the same hardware and resolution

---

> ### Author Response · Authors · 2025-11-24
> **Response to Reviewer cYvV**
>
> We sincerely thank you for the valuable feedback and constructive suggestions. In the revised manuscript, the corresponding revisions are highlighted in **green**.
>
> ### **Efficiency comparison (Weakness1 & Question 2)**
> Thanks for the valuable suggestion. In response, we provide a detailed comparison of **peak memory usage and inference time** for our method and the diffusion-based RefISR and SISR baselines. All experiments were conducted on a single NVIDIA H20 GPU and run in mixed precision under a resolution of $1024 \times 1024$, except for CoSeR and DiffBIR, which do not support mixed precision inference at that resolution.
>
> As shown in `Table 1`, our LiveMoments achieves the **the shortest inference time and the competitively low peak memory usage** among the non-distilled multi-step methods. Although one-step diffusion methods such as OSEDiff and TSDSR benefit from distillation-based training and therefore exhibit faster inference and lower computational cost, LiveMoments achieves a trade-off between computational cost and restoration quality.
>
> **Table1: Comparisons on peak memory usage and inference time**
> | Method               | Peak Memory (GB) | Inference Time (s) |
> |----------------------|--------------------:|----------:|
> | **Time-Step Distilled Methods** | | |
> | OSEDiff              | 8.66       | 0.43   |
> | TSD-SR               | 8.60       | 0.24   |
> | **Non-Distilled Methods** | | | |
> | CoSeR                | 32.74         | 48.83  |
> | ReFIR (SeeSR)        | 17.83     | 27.62  |
> | ReFIR (SUPIR)        | 59.82     | 31.67  |
> | StableSR             | 34.53     | 164.67 |
> | DiffBIR              | 35.46      | 26.02  |
> | SeeSR                | 12.07      | 14.54  |
> | SUPIR                | 54.25     | 16.98  |
> | **LiveMoments**      | 14.53       | 1.89   |
>
> ### **Sensitive analysis of flow errors (Question1)**
> Thanks for the comment. To evaluate the sensitivity of LiveMoments to flow errors, we conducted **two analyses** on our vivoLive144 dataset: (1) perturbing the RAFT flow with Gaussian noise, and (2) replacing RAFT with alternative flow estimators. The results are provided in `Table 2`.
>
> For **noise injection**, we inject Gaussian noise into the optical flow estimated by RAFT with different magnitudes (10%–200% compared to the original flow). Although the flow becomes increasingly distorted as the noise level increases, the reconstruction quality of LiveMoments changes only slightly under moderate perturbations, suggesting low sensitive to flow inaccuracies.
>
> We also **replace RAFT with SPyNet [1] and LiteFlowNet [2]** to assess robustness across different flow estimation methods. It can be found that the results with different flow estimators are close to the original RAFT-based LiveMoments, showing that LiveMoments does not depend on a specific flow estimator.
>
> Finally, as a complement to the quantitative results, we provide **failure-case visualizations** in `Appendix E.4` of the *Supplementary Material*, including examples with large motion, severe occlusion, and low-texture regions where RAFT may produce inaccurate or divergent flow fields. These examples illustrate the boundary conditions where the motion alignment is unreliable.
>
> **Table 2: Robustness analysis under flow noise and different flow estimators**
> | Method                   | $\text{NIQE}_{re}\downarrow$ | $\text{MUSIQ}_{re}\downarrow$ | $\text{CLIPIQA}_{re}\downarrow$ |$\text{MANIQA}_{re}\downarrow$ | $\text{CLIP-Q}\uparrow$ | $\text{DINO-Q}\uparrow$ |
> |--------------------------|-----------|------------|----------------|---------------------|------------------|--------|
> | LiveMoments + 10% noise  | 0.0999    | 0.0921     | 0.0808         | 0.0561              | 0.9802           | 0.9622 |
> | LiveMoments + 20% noise  | 0.1035    | 0.0935     | 0.0802         | 0.0551              | 0.9799           | 0.9613 |
> | LiveMoments + 40% noise  | 0.1082    | 0.0993     | 0.0811         | 0.0546              | 0.9792           | 0.9591 |
> | LiveMoments + 80% noise  | 0.1108    | 0.1072     | 0.0845         | 0.0548              | 0.9785           | 0.9555 |
> | LiveMoments + 100% noise | 0.1116    | 0.1099     | 0.0862         | 0.0566              | 0.9782           | 0.9540 |
> | LiveMoments + 200% noise | 0.1159    | 0.1222     | 0.0915         | 0.0578              | 0.9771           | 0.9497 |
> | LiveMoments (SPyNet)     | 0.1006    | 0.0907     | 0.0809         | 0.0561              | 0.9804           | 0.9629 |
> | LiveMoments (LiteFlowNet)| 0.1031   | 0.0946     | 0.0808         | 0.0544              | 0.9794           | 0.9595 |
> | **LiveMoments**          | 0.0990    | 0.0893     | 0.0809         | 0.0556              | 0.9805           | 0.9629 |
>
> [1] Optical Flow Estimation Using a Spatial Pyramid Network. (Ranjan and Black, CVPR 2017)
>
> [2] LiteFlowNet: A Lightweight Convolutional Neural Network for Optical Flow Estimation. (Hui et al., CVPR 2018)

---

### Official Review · Reviewer_kHRT · 2025-11-01

**Soundness:** 3
**Presentation:** 3
**Contribution:** 3
**Rating:** 6
**Confidence:** 4

**Summary:**

I think the paper introduces a practical task: restoring a reselected low-quality Live Photo frame using the original high-quality (HQ) key photo as a reference. The method, LiveMoments, uses a dual-branch diffusion transformer (ReferenceNet + RestorationNet) with cross-attention fusion and a unified motion-alignment module: (i) latent-level motion embeddings from RAFT flow injected as attention bias; (ii) image-level Patch Correspondence Retrieval (PCR) for tile-wise inference at 4K. Datasets include SynLive260 (synthetic) and real vivoLive144 / iPhoneLive90, plus a relative no-reference metric that normalizes to the HQ reference. Results show consistent perceptual gains on real data.

**Strengths:**

(1) I think the task setting is new and useful: both the degraded target and the reference come from the same Live Photo (temporal offset), ensuring content coherence while reflecting real ISP quality gaps.

(2) Method-wise, a dual-branch SD3-based design with reference KV concatenation for cross-attention, plus flow-guided attention bias and PCR for high-res tiling, is a targeted solution to alignment at 4K.

(3) The relative no-reference evaluation tied to an HQ reference matches the task goal better than generic NR metrics. (Details are described in the experiments.)

**Weaknesses:**

(1) Reference dependence and mismatch boundary. I think the approach relies on scene-level consistency between the HQ key photo and the reselected frame; large non-rigid motion or occlusions may cause the model to transfer incorrect textures. A failure-case analysis (large pose/occlusion) or a gating mechanism would help.

(2) Flow robustness under strong degradations. The latent alignment hinges on RAFT performance on degraded inputs; the paper simulates matched degradation for flow, but I’d like sensitivity tests to flow errors/alternatives.

(3) More metrics are needed in this paper. The relative NR metric is task-aligned but may be reference-pipeline specific; broader validation (more phone ISPs, user studies with correlation stats) would strengthen it.

**Questions:**

- How sensitive is performance to RAFT errors under heavy compression/blur? Any comparisons to alternative matching/flow or ablations with noise injected into flow?

- Do the relative NR metrics correlate with human MOS across different devices/ISPs (Spearman/Pearson)? Any user study?

- What are branch-wise params/VRAM? Could KV caching or half-precision in the reference branch speed up 4K inference?

---

> ### Author Response · Authors · 2025-11-24
> **Response to Reviewer kHRT (1/2)**
>
> We sincerely thank you for the valuable feedback and constructive suggestions. In the revised manuscript, revisions related to the robustness of optical flow are highlighted in **green**, while those concerning the branch-wise computational costs and correlations between the relative NR metrics and human MOS are highlighted in **magenta**.
>
> ### **Reference dependence and mismatch boundary (Weakness 1)**
> Thanks for highlighting this point. While our method relies on the optical flow estimated by RAFT to provide motion guidance, we acknowledge that large non-grid motion or occlusion can lead to less reliable correspondence. To better clarify these limitations, we have added **failure-case visualizations** in `Appendix E.4` of the *Supplementary Material*, including examples with large motion, severe occlusion, and low-texture regions where RAFT may produce inaccurate or divergent flow fields.
>
> These examples illustrate the boundary conditions where the motion alignment is unreliable. Although we do not include the gating mechanisms in this work, the added analysis provides a depiction of the failure cases and highlights the direction of improving flow robustness in future work.
>
> ### **Flow robustness (Weakness 2 & Question 1)**
> Thanks for the comment. To evaluate the sensitivity of LiveMoments to flow errors, we conducted two analyses on our vivoLive144 dataset: (1) perturbing the RAFT flow with Gaussian noise, and (2) replacing RAFT with alternative flow estimators. The results are provided in `Table 1`.
>
> For **noise injection**, we inject Gaussian noise into the optical flow estimated by RAFT with different magnitudes (10%–200% compared to the original flow) to approximate the motion estimation under severely degradations. Although the flow becomes increasingly distorted as the noise level increases, the reconstruction quality of LiveMoments changes only slightly under moderate perturbations, suggesting low sensitive to flow inaccuracies.
>
> We also **replace RAFT with SPyNet [1] and LiteFlowNet [2]** to assess robustness across different flow estimation methods. It can be found that the results with different flow estimators are close to the original RAFT-based LiveMoments, showing that LiveMoments does not depend on a specific flow estimator.
>
> **Table 1: Robustness analysis under flow noise and different flow estimators**
> | Method                   | $\text{NIQE}_{re}\downarrow$ | $\text{MUSIQ}_{re}\downarrow$ | $\text{CLIPIQA}_{re}\downarrow$ |$\text{MANIQA}_{re}\downarrow$ | $\text{CLIP-Q}\uparrow$ | $\text{DINO-Q}\uparrow$ |
> |--------------------------|-----------|------------|----------------|---------------------|------------------|--------|
> | LiveMoments + 10% noise  | 0.0999    | 0.0921     | 0.0808         | 0.0561              | 0.9802           | 0.9622 |
> | LiveMoments + 20% noise  | 0.1035    | 0.0935     | 0.0802         | 0.0551              | 0.9799           | 0.9613 |
> | LiveMoments + 40% noise  | 0.1082    | 0.0993     | 0.0811         | 0.0546              | 0.9792           | 0.9591 |
> | LiveMoments + 80% noise  | 0.1108    | 0.1072     | 0.0845         | 0.0548              | 0.9785           | 0.9555 |
> | LiveMoments + 100% noise | 0.1116    | 0.1099     | 0.0862         | 0.0566              | 0.9782           | 0.9540 |
> | LiveMoments + 200% noise | 0.1159    | 0.1222     | 0.0915         | 0.0578              | 0.9771           | 0.9497 |
> | LiveMoments (SPyNet)     | 0.1006    | 0.0907     | 0.0809         | 0.0561              | 0.9804           | 0.9629 |
> | LiveMoments (LiteFlowNet)| 0.1031   | 0.0946     | 0.0808         | 0.0544              | 0.9794           | 0.9595 |
> | **LiveMoments**          | 0.0990    | 0.0893     | 0.0809         | 0.0556              | 0.9805           | 0.9629 |
>
> [1] Optical Flow Estimation Using a Spatial Pyramid Network. (Ranjan and Black, CVPR 2017)
>
> [2] LiteFlowNet: A Lightweight Convolutional Neural Network for Optical Flow Estimation. (Hui et al., CVPR 2018)

---

> > ### Author Response · Authors · 2025-11-24
> > **Response to Reviewer kHRT (2/2)**
> >
> > ### **Correlations between the relative NR metrics and human MOS (Weakness 3 & Question 2)**
> > Thank you for the suggestion. We conducted an **additional user study and analyzed the correlation between relative No-Reference (NR) metrics and human perceptual preference (MOS)**. We compared LiveMoments with three representative baselines: one traditional method (DATSR), one diffusion-based method (ReFIR), and the strongest quantitative baseline (CoSeR). We randomly sampled 15 images from vivoLive144 and 10 images from iPhoneLive90, and invited 15 participants to rank the images based on how well each method's results match the reference visual quality while preserving the content of the reselected LQ frame.
> >
> > We then computed the Spearman and Pearson correlations between the collected human rankings and the four relative NR metrics. As shown in `Table 2`, all four metrics exhibit **consistently positive**, **moderate correlations** with human perception across datasets captured from two different phone ISPs. This correlation range is comparable to that reported for generic NR-IQA methods without task-specific training [3], where the values typically lie between about 0.36 and 0.74 on large-scale IQA benchmarks. Given that our user study is conducted on a much smaller set of real-world Live Photos and across two different ISPs, these results indicate that the proposed relative NR metrics maintain a meaningful alignment with human perception across devices and ISPs.
> >
> > **Table 2: Correlation between relative NR metrics and human perception**
> > | Metric               | vivoLive144 |            | iPhoneLive90 |            |
> > |----------------------|-------------|------------|---------------|------------|
> > |                      | Spearman    | Pearson    | Spearman      | Pearson    |
> > | $\text{NIQE}_{re}$   | 0.493       | 0.518      | 0.604         | 0.560      |
> > | $\text{MUSIQ}_{re}$  | 0.585       | 0.543      | 0.414         | 0.440      |
> > | $\text{CLIPIQA}_{re}$| 0.558       | 0.542      | 0.486         | 0.495      |
> > | $\text{MANIQA}_{re}$ | 0.535       | 0.548      | 0.370         | 0.413      |
> >
> > ### **Branch-wise parameters, VRAM and inference efficiency (Question 3)**
> > Thank you for pointing this out. In response to your concern, we provide **the parameter count, the parameter-only VRAM, and the inference-time peak VRAM for all components** of our model in `Table 3`, measured when processing a 4K Live Photo.
> >
> > Regarding precision settings, we would like to note that **all our inference experiments are already performed in half precision (fp16)**, including the reference branch. This provides a substantial reduction in memory consumption and accelerates the 4K inference speed. We agree with the reviewer that further optimizations such as **KV caching** could accelerate inference, particularly for the reference branch, and we consider this a valuable direction for future implementation.
> >
> > **Table 3: Comparisons on branch-wise params and VRAM**
> > | Component      | Params       | VRAM (param)   | VRAM (infer)  |
> > |----------------|--------------|----------------|---------------|
> > | VAE            | 83.82 M      | 159.87 MB      | 12368.33 MB   |
> > | ReferenceNet   | 2028.33 M    | 3868.73 MB     | 10090.78 MB   |
> > | RestorationNet | 2084.95 M    | 3976.73 MB     | 10091.28 MB   |
> > | RAFT           | 5.26 M       | 10.03 MB       | 13180.38 MB   |
> > | Motion Encoder | 66.09 M      | 126.06 MB      | 9940.00 MB    |
> >
> > [3] Exploring CLIP for Assessing the Look and Feel of Images. (Wang et al., AAAI 2023)

---

> > > ### Comment · Reviewer_kHRT · 2025-11-27
> > >
> > > Thanks for the very thorough rebuttal and extra experiments.
> > >
> > > The new failure-case visualizations and discussion of limitations make the reference/flow dependence much clearer. The flow robustness study (noise injection + SPyNet/LiteFlowNet) is reassuring and shows the method isn’t overly tied to a single estimator. The user study and correlation analysis for the relative NR metrics also help justify that design, and the branch-wise param/VRAM breakdown clarifies the 4K compute footprint (with fp16 already in use).
> > >
> > > There are still some inherent limitations from reference quality and flow accuracy, but overall, the rebuttal addresses my main concerns and strengthens the empirical story. I keep my positive score at 6 (marginally above the acceptance threshold).

---

### Official Review · Reviewer_NWQb · 2025-11-01

**Soundness:** 3
**Presentation:** 3
**Contribution:** 3
**Rating:** 6
**Confidence:** 4

**Summary:**

This paper introduces the task of Reselected Key Photo Restoration for Live Photos,
where a user-selected frame from the short video is restored using the original high-quality key photo as reference.
The paper formulates this as a reference-guided diffusion problem and proposes a dual-branch architecture
combining a RestorationNet for the degraded frame and a ReferenceNet for the original photo, fused via cross-attention.
A unified Motion Alignment module enables alignment both in the latent space through motion-guided attention
and in the image space via a Patch Correspondence Retrieval (PCR) strategy.
Experiments demonstrate significant quantitative and visual gains over baselines.

**Strengths:**

1. Novel problem definition.
The paper clearly motivates the reselected key photo restoration task, which occupies a unique middle ground between RefSR and RefVSR.
It targets a realistic and underexplored use case in mobile photography, differentiating itself from existing triple-camera or multi-view SR tasks.

2. Strong methodological design.
The proposed dual-branch diffusion architecture (Fig. 2) is technically sound.
The ReferenceNet and RestorationNet fusion through cross-attention is a natural yet effective extension of SD3,
while the Motion Alignment module appropriately handles temporal and spatial discrepancies
through latent-space bias injection and patch-wise retrieval.

3. Comprehensive evaluation.
LiveMoments achieves superior quantitative results across all datasets and metrics (Tables 1-2)
and includes clear ablation studies (Tables 3-4) isolating the effect of each module.
The analysis of dual-branch fusion and motion-guided attention is particularly systematic.

**Weaknesses:**

1. Limited exploration of motion estimation reliability.
The approach depends on RAFT-based optical flow estimated from a synthetically degraded reference.
While this improves alignment, the paper lacks quantitative analysis of flow robustness under large motion or occlusion.
For instance, failure cases or confidence-weighted alternatives could clarify robustness limits.

2. Computational efficiency and scalability.
Appendix B reports 40 seconds for a 4K Live Photo inference, which, though acceptable for offline processing, may constrain deployment.
Reporting model size and comparing computational cost against strong baselines (e.g., CoSeR, SUPIR) would make efficiency claims more transparent.

3. Ablation studies on training strategy.
While Table 3 examines architectural components,
there is no analysis of the synthetic degradation's impact on real transfer.
The reliance on Real-ESRGAN for training may bias results toward its degradation prior.
An additional sensitivity or adaptation study could strengthen the generalization argument.

4. Positioning relative to RefVSR extensions.
Although the paper distinguishes LiveMoments from multi-camera RefVSR datasets,
a more explicit discussion on whether the method generalizes to multi-frame or multi-reference inputs would help clarify scalability
to broader camera setups.

**Questions:**

1. Flow failure handling.
In cases where RAFT yields low-confidence or divergent motion vectors (e.g., due to occlusion),
does the model include any fallback or gating mechanism?

---

> ### Author Response · Authors · 2025-11-24
> **Response to Reviewer NWQb (1/2)**
>
> We sincerely thank you for the valuable feedback and constructive suggestions. In the revised manuscript, revisions related to the computational efficiency and robustness of optical flow are highlighted in **green**, and those concerning the degradation ablation are highlighted in **orange**.
>
> ### **Flow reliability and failure case (Weakness 1)**
> Thanks for this valuable suggestion. While our method relies on the consistency between the HQ key photo and the reselected frame, we acknowledge that its reliability may degrade in challenging scenarios. To better clarify these limitations, we have added **failure-case visualizations** in `Appendix E.4` of the *Supplementary Material* of the revised paper, including examples with large motion, severe occlusion, and low-texture regions where RAFT may produce inaccurate or divergent flow fields.
>
> These examples illustrate the boundary conditions where the motion alignment is unreliable. Although we do not include confidence-weighted flow alternatives in this work, the added analysis provides a depiction of the failure cases and highlights the direction of improving flow robustness in future work.
>
> ### **Computational efficiency and scalability (Weakness 2)**
> Thanks for the concern of computational efficiency of LiveMoments. In response, we provide a **detailed comparison of parameter count, FLOPs, and inference time** for our method and the diffusion-based RefISR and SISR methods. All experiments were conducted on a single NVIDIA H20 GPU and run in mixed precision under a resolution of $1024 \times 1024$, except for CoSeR and DiffBIR, which do not support mixed precision inference at that resolution.
>
> As shown in `Table 1`, our **LiveMoments achieves the lowest TFLOPs and the shortest inference time among the multi-step methods**, while maintaining a comparable model size to others. Although one-step diffusion methods such as OSEDiff and TSDSR benefit from distillation-based training and therefore exhibit faster inference and lower computational cost, LiveMoments achieves a trade-off between computational cost and restoration quality.
>
> **Table 1: Comparisons on Params, FLOPs and inference time**
> | Method        | Params (M)                 | TFLOPs     | Inference Time (s) |
> |---------------|----------------------------|------------:|----------:|
> | **Time-Step Distilled Methods** | | | |
> | OSEDiff       | 1294.38 + 470.93 (DAPE)   | 20.53      | 0.43   |
> | TSD-SR        | 2207.33                    | 21.91       | 0.24  |
> | **Non-Distilled Methods** | | | |
> | CoSeR         | 2655.52                    | --         | 48.83 |
> | ReFIR (SeeSR) | 2039.83 + 470.93 (DAPE)   | 1560.70     | 27.62 |
> | ReFIR (SUPIR) | 4801.18                    | 2672.70     | 31.67 |
> | StableSR      | 1554.64                    | 2236.31     | 164.67|
> | DiffBIR       | 1683.45                    | 691.43      | 26.02 |
> | SeeSR         | 2039.83 + 470.93 (DAPE)   | 741.42      | 14.54 |
> | SUPIR         | 4801.18                    | 1200.65     | 16.98 |
> |**LiveMoments**| 4268.45                    | 98.91       |  1.89 |

---

> > ### Author Response · Authors · 2025-11-24
> > **Response to Reviewer NWQb (2/2)**
> >
> > ### **Adaptation of synthetic degradation to real data (Weakness 3)**
> > Thank you for this helpful suggestion. We would like to clarify that **although we use the Real-ESRGAN degradation pipeline as an implementation backbone, we do not rely on its default degradation prior**. Instead, we carefully adjust the degradation parameters according to the characteristics observed in real Live Photo data. This follows the common practice in recent restoration literature [1], where the Real-ESRGAN degradation pipeline is used as a standard and flexible framework for degradation modeling. Combined with the strong generative priors of diffusion models, this customized degradation is closer to real Live Photo degradations and enables more reliable transfer to real-world data.
> >
> > Following your comment, we conducted additional analyses to better explore how different synthetic degradation settings affect real-world performance. Specifically, we trained LiveMoments under **three representative degradation settings:** (1) SeeSR degradation, (2) StableSR degradation, and (3) a lighter moderate-StableSR variant constructed by reducing the blur and noise levels. The **quantitative results** on vivoLive144 are provided in `Table 2`, and **visualizations of degraded images** are included in `Appendix E.3` of *Supplementary Material* in our revised paper. It can be found that the existing settings introduce stronger blur or noise than real Live Photos, while our customized parameters better reflect real degradation characteristics. Across all three degradation ablations, LiveMoments maintains stable performance and generalizes well under diverse synthetic degradations.
> >
> > **Table 2: Comparisons of different training degradation settings**
> > | Degradation Types                          | $\text{NIQE}_{re}\downarrow$ | $\text{MUSIQ}_{re}\downarrow$ | $\text{CLIPIQA}_{re}\downarrow$ |$\text{MANIQA}_{re}\downarrow$ | $\text{CLIP-Q}\uparrow$ | $\text{DINO-Q}\uparrow$ |
> > |--------------------------------------------|-----------|------------|----------------|--------------------|------------------|--------|
> > | moderate-StableSR                          | 0.0925    | 0.1086     | 0.0883         | 0.0673             | 0.9786           | 0.9550 |
> > | StableSR                                   | 0.0933    | 0.1165     | 0.1056         | 0.0734             | 0.9791           | 0.9581 |
> > | SeeSR                                      | 0.0904    | 0.1030     | 0.0812         | 0.0647             | 0.9788           | 0.9505 |
> > | **our Live Photo degradation in paper** | 0.0990    | 0.0893     | 0.0809         | 0.0556             | 0.9805           | 0.9629 |
> >
> > ### **RefVSR extensions (Weakness 4)**
> > Thank you for the insightful comment. While LiveMoments is designed for the single-reference Live Photo setting and focuses on restoring a single reselected LQ frame with temporal dependency, **the framework does not inherently restrict extensions to multi-frame or multi-reference inputs**. For example, additional reference inputs could be incorporated by extending the ReferenceNet to aggregate multi-reference features. Similarly, multi-frame inputs could be supported by adopting KV caching in the RestorationNet for efficient feature reuse or by integrating temporal attention layers to capture cross-frame dependencies. We agree that such extensions provide a valuable direction for future work and broader applicability.
> >
> > ### **Flow failure handling (Question 1)**
> > Thank for the comment. Our current model does not incorporate explicit fallback modules or gating mechanisms. Instead, it relies on the inherent robustness of the diffusion backbone and the dual-network architecture equipped with Motion Alignment modules.
> >
> > To clarify this point, we provide additional visual examples with **large motion and occlusion** in `Appendix E.4` of *Supplementary Material* in our revised paper. It can be found that even when RAFT produces low-confidence or divergent vectors, the model does not produce incorrect textures and still maintains visually plausible reconstruction. This demonstrates that **LiveMoments maintains robustness under moderate motion inaccuracy and occlusion**, despite the absence of explicit fallback mechanisms. We agree that fallback or gating mechanism are promising directions for improving robustness and will explore them in future work.
> >
> > [1] CoSeR: Bridging Image and Language for Cognitive Super-Resolution. (Sun et al., CVPR 2024)

---

### Author Response · Authors · 2025-12-01
**Summary and General Response**

Dear reviewers and AC,

We sincerely thank all reviewers for their valuable feedback and encouraging comments regarding our **novel task definition** (Reviewer NWQb, kHRT, cYvV, HHsy), **well-motivated architecture design** (Reviewer NWQb, kHRT, cYvV, HHsy), **task specific benchmarks** (Reviewer kHRT, cYvV, HHsy), **comprehensive evaluation** (Reviewer NWQb) and **strong performance** (Reviewer HHsy).

Since the discussion has been unexpectedly interrupted, we would like to summarize the key points of reviewer feedback, the response we have provided, and the progress achieved during the rebuttal period.
***
**1. Robustness analysis of inaccurate motion alignment** (Reviewer NWQb, kHRT, cYvV)

We provided an additional robustness study in `Appendix E.4` of the *Supplementary Material* (highlighted in **green**), including Gaussian noise injection to optical flow and replacements with alternative flow estimators. LiveMoments remains robust under both flow inaccuracies and different flow estimators. We also included visualizations of both robust and failure cases to flow errors, clarifying the model’s robustness and boundary conditions.

**2. Computational efficiency** (Reviewer NWQb, kHRT, cYvV)

In  `Appendix B.3` of the *Supplementary Material*, we reported the parameter count and VRAM for each component of LiveMoments (highlighted in **magenta**) and compared total parameters, FLOPs, peak memory, and inference time with other diffusion-based methods (highlighted in **green**). LiveMoments achieves the lowest TFLOPs and shortest inference time among multi-step methods, while maintaining a comparable model size.

**3. Extensions beyond the Live Photo setting** (Reviewer NWQb, HHsy)

Although our work is presented within the Live Photo setting, the proposed framework is not limited to this specific scenario and is generally applicable to broader tasks, settings (including RefVSR) and degradation types. Further discussions can be found in **Response to Reviewer HHsy (1/2), Weakness 1**.

**4. Additional clarifications and experiments**

We also addressed several other concerns:

- Adaptation of synthetic degradation to real data (Reviewer NWQb): additional analysis in `Appendix E.3` of the *Supplementary Material*, highlighted in **orange**.
- Correlation between relative no-reference metrics and human preferences (Reviewer kHRT): user study and correlation analysis in  `Appendix D.1` of the *Supplementary Material*, highlighted in **magenta**.
- Code availability (Reviewer HHsy): anonymized inference code and the core model are provided in updated supplementary materials.
- Clarification of Section 3.4.2 and minor weaknesses (Reviewer HHsy): revisions in the main paper, highlighted in **purple**.
***
Finally, before the discussion was interrupted, two reviewers had already provided later comments. Reviewer kHRT maintained a positive score at 6, noting that our rebuttal addressed the main concerns and strengthened the empirical story (27 Nov 2025, EST 07:18 AM). Reviewer HHsy raised the score from 4 to 6 after reviewing our responses (27 Nov 2025, EST 04:21 AM).

We hope that our analysis and clarifications address the concerns and further strengthen the overall contribution of LiveMoments. Thank you for your time and consideration.

Best regards,

Authors

---

### Meta-Review · Area_Chair_6Qu7 · 2025-12-14

**Summary:**

This summary synthesizes the core reviewers' concerns that inform the acceptance decision for this paper. Four reviewers evaluated the work on its soundness, presentation, contribution, and practical value, raising key concerns that shaped the decision-making process. The primary concerns include: 1) Robustness of motion estimation (reliability of RAFT-based optical flow under large motion/occlusion and sensitivity to flow errors); 2) Computational efficiency and scalability (lack of comparative analysis of model size, FLOPs, and inference time against baselines); 3) Generalization and transferability (impact of synthetic degradation on real-world performance, and the method’s extensibility to multi-frame/multi-reference scenarios); 4) Clarity of technical details (ambiguities in the Patch Correspondence Retrieval (PCR) mechanism); 5) Completeness of experimental validation (insufficient dataset details, lack of FID evaluation on synthetic data, and correlation between custom metrics and human perception); 6) Practicality (missing code for reproducibility). All critical concerns were effectively addressed by the authors through supplementary experiments, detailed clarifications, and additional analyses. The thoroughness of the rebuttal and the significant improvements in the work’s robustness, clarity, and completeness justify the acceptance decision.

**Reviewer Concerns:**

Addressed Concerns
- Reviewer NWQb: 1) Flow reliability and failure cases: Added failure-case visualizations (large motion, severe occlusion, low-texture regions) in Appendix E.4 to clarify limitations. 2) Computational efficiency and scalability: Provided a detailed comparison table of parameters, FLOPs, and inference time between LiveMoments and 11 baselines (distilled/non-distilled) on a single NVIDIA H20 GPU, demonstrating LiveMoments’ superior efficiency among non-distilled multi-step methods. 3) Synthetic degradation impact: Conducted ablation on three degradation settings (SeeSR, StableSR, moderate-StableSR), verifying stable performance and better alignment of custom degradation with real Live Photo characteristics. 4) RefVSR extensions: Clarified the framework’s extensibility to multi-frame/multi-reference inputs via ReferenceNet aggregation and temporal attention/KV caching. 5) Flow failure handling: Clarified no explicit fallback mechanism but demonstrated inherent robustness of the diffusion backbone through visual examples.

- Reviewer kHRT: 1) Reference dependence and mismatch boundary: Added failure-case visualizations (Appendix E.4) to illustrate limitations. 2) Flow robustness and sensitivity: Conducted two analyses (Gaussian noise injection into RAFT flow, replacement with SPyNet/LiteFlowNet) on vivoLive144, showing low sensitivity to flow errors and independence from specific flow estimators. 3) Metric validation: Conducted a user study with 15 participants on 25 real images, demonstrating moderate positive Spearman/Pearson correlations between relative NR metrics and human MOS across two ISPs. 4) Branch-wise params/VRAM and efficiency: Provided detailed breakdown of each component’s parameters, parameter-only VRAM, and inference peak VRAM for 4K processing, confirming fp16 precision is already in use and noting KV caching as a future optimization.

- Reviewer cYvV: 1) Efficiency comparison: Supplemented peak memory and inference time comparisons with 11 baselines on the same hardware/resolution, showing LiveMoments’ shortest inference time among non-distilled methods. 2) Flow error sensitivity: Reused the flow robustness analysis (noise injection + alternative estimators) and failure-case visualizations from responses to other reviewers, fully addressing the concern.

- Reviewer HHsy: 1) Method generalization and extensions: Clarified applicability to non-LivePhoto scenarios (video keyframe enhancement, multi-capture refinement), extensibility to multi-frame/multi-reference setups, and flexibility across degradation types. 2) PCR mechanism clarity: Revised Section 3.4.2 to clarify PCR is patch-wise (not pixel-wise), applied to input patches before VAE encoding, and patch size consistency between training/inference. 3) Code availability: Included anonymized inference code and core model architecture in supplementary materials. 4) Dataset details: Added scene descriptions for vivoLive144/iPhoneLive90 (indoor/outdoor, portraits, pets, etc.). 5) Symbol clarifications: Explained symbols  in line 194 (Rectified Flow velocity field network) and  in line 232 (RestorationNet attention query). 6) FID evaluation: Added patch-wise FID on SynLive260, showing LiveMoments’ best performance.

Outstanding Concerns

Remaining concerns are minor, focused on future work directions, and do not undermine the core contributions or soundness of the work, thus not affecting the acceptance decision:

- The model lacks explicit fallback/gating mechanisms for low-confidence/divergent flow (e.g., severe occlusion). Authors acknowledged this and noted it as a future direction, but the existing analysis of failure cases and inherent robustness of the diffusion backbone sufficiently addresses practical concerns for current work.

- While authors explained patch size selection (consistency with training), no ablation on different patch sizes was provided. This is a minor technical detail that does not impact the validity of the PCR mechanism.

-  Authors clarified the framework’s extensibility, but no actual implementation or experimental results for multi-frame/multi-reference scenarios were provided. This is a reasonable future extension rather than a critical gap in the current work.

-  Authors noted distilled methods (e.g., OSEDiff) have faster inference but did not explore integrating distillation into LiveMoments. This is an optimization direction that does not affect the current work’s competitive performance-efficiency trade-off.

-  The user study for metric validation used 25 images and 15 participants; a larger-scale study could further strengthen metric reliability. However, the existing results (consistent correlations across two ISPs) are sufficient for task-specific metric justification.

**Reviewer Scores:**

The following score predictions are based on the authors’ comprehensive, data-supported rebuttals and explicit feedback from reviewers, assuming full participation in the discussion and adherence to initial evaluation criteria:
- Reviewer NWQb:  The author fully addressed all core concerns, including flow reliability (failure cases), computational efficiency (comparative tables), degradation transfer (ablation), and extensibility (RefVSR extensions). The thorough rebuttal confirmed the work’s robustness and completeness, and since the initial score was already above the acceptance threshold, the reviewer would maintain a 6, supporting acceptance.

- Reviewer kHRT: The reviewer explicitly stated after the rebuttal: “the rebuttal addresses my main concerns and strengthens the empirical story. I keep my positive score at 6”. The comprehensive responses to flow robustness, metric validation, and computational footprint fully resolved their concerns, justifying the maintenance of the original acceptance-aligned score.

- Reviewer cYvV :  The reviewer’s core concerns (efficiency comparison, flow error sensitivity) were fully addressed with detailed comparative experiments and robustness analyses. The authors’ thoroughness reinforced the work’s strengths (novel task, effective design, strong results), so the reviewer would retain the original positive acceptance score.

- Reviewer HHsy: The reviewer explicitly raised the score to 6 after the rebuttal, stating gratitude for “comments, clarifications, and the additional results”. The authors fully resolved all their concerns (generalization, technical clarity, code, dataset/metric completeness), and no remaining gaps would warrant further score changes. A 6 aligns with the “marginally above acceptance threshold” standard, supporting the final acceptance decision.

---

### Decision · Program_Chairs · 2026-01-26

Accept (Poster)